# Cardio-audio synchronization elicits neural and cardiac surprise responses in human wakefulness and sleep
Andria Pelentritou [1] ✉, Christian Pfeiffer[2], Sophie Schwartz[3,4] & Marzia De Lucia [1] ✉

The human brain can encode auditory regularities with fixed sound-to-sound intervals and with sound onsets locked to cardiac inputs. Here, we investigated auditory and cardio-audio regularity encoding during sleep, when bodily and environmental stimulus processing may be altered. Using electroencephalography and electrocardiography in healthy volunteers ($N = 26$) during wakefulness and sleep, we measured the response to unexpected sound omissions within three regularity conditions: synchronous, where sound and heartbeat are temporally coupled, isochronous, with fixed sound-to-sound intervals, and a control condition without regularity. Cardio-audio regularity encoding manifested as a heartbeat deceleration upon omissions across vigilance states. The synchronous and isochronous sequences induced a modulation of the omission-evoked neural response in wakefulness and N2 sleep, the former accompanied by background oscillatory activity reorganization. The violation of cardio-audio and auditory regularity elicits cardiac and neural responses across vigilance states, laying the ground for similar investigations in altered consciousness states such as coma and anaesthesia.

The processing of auditory regularity is a basic brain mechanism that enables the rapid detection of unexpected stimuli which can persist even in altered states of consciousness such as sleep and coma[1]. The mechanism underlying auditory regularity encoding has mostly been studied by investigating the neural responses to deviant sounds interrupting a sequence of repeated standard stimuli as reported in healthy human wakefulness[2,3], during sleep[4–6], and in disorders of consciousness patients[7–9]. This rudimental component of auditory discrimination, known as mismatch negativity (MMN), has often been interpreted in the framework of the predictive coding theory[2,3,10,11]. According to this theory, the MMN may arise from the contribution of multiple, non-exclusive mechanisms including repetition suppression in response to frequent stimuli and generation of a 'prediction error' following an unexpected mismatch between the predicted and presented stimuli. The predictive nature of the neural responses to violations within regular auditory sequences has received experimental support from studies on unexpected omissions[12–16] wherein the top-down prediction is not confounded by the neural response to deviant sound stimuli.

In addition to external stimuli, the human brain receives internally generated bodily signals, which constitutes a continuous source of sensory inputs. Simultaneous environmental and bodily information may compete for neural resources and influence their respective processing[17]. Accordingly, previous studies have shown that bodily signals and their associated neural representation modulate perception and cognition[18–20]. In particular, the neural processing of cardiac signals, as measured by heartbeat evoked potentials (HEPs[21]), may determine whether a stimulus is consciously perceived[22], influence emotional processing[23–25], and could account for the first person perspective in perceptual experience[26]. In addition, heartbeat processing can be used to measure interoceptive ability (i.e., the ability of sensing the inner bodily state), which in turn can influence affective control, physical and mental wellbeing in health and a variety of clinical conditions[27,28].

In this context, it is also plausible that the rhythmic information from the heartbeat could modulate the processing of temporally-organized sequences of external sensory signals. Recent experimental paradigms have demonstrated that auditory sequences locked to the ongoing heartbeat generate an auditory temporal prediction even in the absence of fixed sound-to-sound intervals[29–31]. Whereas such prediction across interoceptive and exteroceptive signals may arise during the awake state, it is unclear whether it may be observed during sleep, when the processing of external information and its temporal structure is reduced compared to wakefulness[32]. Previous studies in sleep have investigated interoceptive and exteroceptive stimulus

¹Laboratoire de Recherche en Neuroimagerie (LREN), Lausanne University Hospital and University of Lausanne, 1011 Lausanne, Switzerland. ²Robotics and Perception Group, University of Zurich, 8050 Zurich, Switzerland. ³Department of Neuroscience, Faculty of Medicine, University of Geneva, 1211 Geneva, Switzerland. ⁴Swiss Center for Affective Sciences, University of Geneva, 1202 Geneva, Switzerland. ✉e-mail: Antria.Pelentritou@chuv.ch; Marzia.De-Lucia@chuv.ch

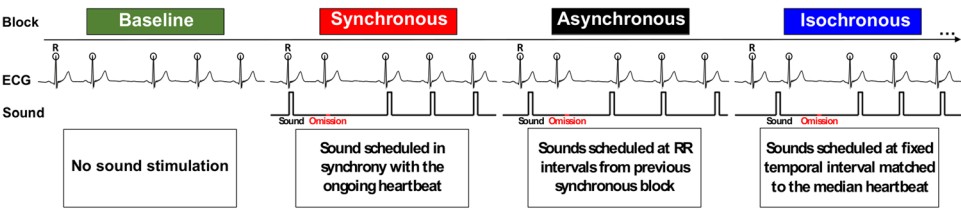

**Fig. 1 | Experimental paradigm overview.** The data acquisition was organized in four block types, administered in pseudo-random order. No auditory stimulation was performed during the Baseline condition. In the Synchronous condition, the R peaks (R; circles) were detected online in the ECG signal and the sounds were administered at a fixed 52 ms R-to-sound. In the Asynchronous condition, sound-to-sound intervals from a preceding Synchronous blocks were administered resulting in variable R-to-sound and sound-to-sound intervals. In the Isochronous condition, the median R-to-R interval from a preceding Synchronous block was computed and served as the fixed sound-to-sound interval. Sound omissions occurred pseudo-randomly for 20% of trials within each block.

processing independently. First, heartbeat stimulus processing across sleep stages[33–35] and its modulation in patients with sleep disorders in comparison to healthy controls[36,37] suggest some preservation of interoceptive signal processing in sleep. Second, evidence of rudimental sensory stimulus processing in sleep is documented in a large body of literature (e.g.,[6,38–40]).

Here, we investigated whether the heartbeat and auditory signals are integrated and inform auditory regularity encoding. Addressing this question is important in light of previous evidence showing that the neural processing of cardiac signals can boost the integration of conscious percepts[21,22], while the potentially beneficial effect of the heartbeat on external stimulus processing during sleep, when conscious awareness of the external environment is at its lowest, has, to the best of our knowledge, never been shown. Specifically, we investigated whether detecting regularity in trains of auditory stimuli may benefit from their temporal alignment with the ongoing heartbeat in sleep and wakefulness. We hypothesized that, as the brain gradually disconnects from the environment during sleep, bodily signals may become increasingly critical for informing auditory regularity encoding and the detection of unexpected violation of such regularities. Healthy volunteers underwent two separate sessions of simultaneous electrocardiography (ECG) and electroencephalography (EEG) recordings during wakefulness and full night sleep. Participants passively listened to two possible varieties of auditory regularities. In the first condition, sounds were presented at a fixed short delay relative to the ongoing heartbeat (synchronous or synch condition), in the second, sounds were presented at a fixed sound-to-sound interval (isochronous or isoch condition) and a third, where sounds were presented without any specific regularity (asynchronous or asynch condition) served as a control condition (Fig. 1). To assess the preservation of regularity encoding in all conditions, we measured the cardiac responses (from ECG) and the neural responses (from EEG) to unexpected omissions interspersed within the auditory sequences. Upon sound omissions in wakefulness and sleep, we expected prediction error generation in both the synch and isoch conditions as a consequence of the cardio-audio regularity and auditory regularity encoding. As both the cardiac and the neural responses during sound omission may carry the signature of this prediction error generation[30,41], we investigated the ECG and EEG responses to unexpected omission across different conditions of temporal alignments between cardiac and auditory signals.

## Results

Below, we first provide a general description of the obtained dataset ('Participants and sleep characteristics'). Next, we report the cardiac (i.e., ECG) response to omissions which elucidates whether the heartbeat is modulated by the omission of a sound, across all auditory regularity conditions and vigilance states ('Cardiac omission response'). In the 'Neural omission response' section, we report the neural (i.e., EEG) response to omissions during wakefulness and N2 sleep, which evidences omission-related neural responses when auditory regularity is based on a fixed relationship between cardiac stimuli and sound onset (synch condition), as well as with fixed sound-to-sound interval (isoch condition). In the 'Slow oscillation analysis' section, we explore how different types of auditory regularities affect the background slow oscillations (SOs) in N2 sleep, as one potential mechanism

by which the sleeping brain might build predictions based on the temporal relationship between auditory stimuli. Finally, the 'Quality control analysis' section summarizes the results of the control analyses aiming at verifying the efficacy of the experimental manipulation. In particular, this last section establishes the validity of our interpretation of the ECG and EEG results by demonstrating that these results are unlikely to be explained by other factors that may differ between experimental conditions.

Note that in the following, R refers to the R peak of the ECG waveform, S represents the sound onset, RR the R peak-to-R peak time interval, RS the R peak-to-sound onset time interval, SR the sound to the next R peak time interval, SS the sound-to-sound time interval, and that the variability of these variables is calculated as the standard error of the mean (SEM).

### Participants and sleep characteristics

Twenty-six healthy volunteers participated in the study (14 female; 1 left-handed; mean age: 27 years, range: 20–35 years) and each took part in a wakefulness and sleep recording session. All 26 participants were included for wakefulness while the sleep dataset included 25 participants (13 female; 1 left-handed; mean age: 26 years, range: 20–35 years) due to malfunctioning equipment during the sleep session for 1 participant. The sleep characteristics recorded during the experimental night across the eligible population ($N = 25$) are summarized in Table 1.

### Cardiac omission response

We investigated whether regularity violation upon omission of expected sounds could elicit a cardiac response across vigilance states. We analyzed heartbeat changes based on the RR intervals extracted from the ECG in response to sound omissions as a function of the auditory conditions. In particular, we extracted the RR intervals prior, during, and two intervals after omission, $RR_{-1}$, $RR_{om}$, $RR_{+1}$, $RR_{+2}$, respectively. RR intervals were normalized (by subject-wise division of each of the investigated average RR intervals by the average $RR_{-1}$) so that the results would not be attributable to inter-subject variability in RR intervals (Fig. 2). We included all 26 participants for the wakefulness session, 15 for N1 sleep, 24 for N2 sleep, 18 for N3 sleep, and 15 for REM sleep (Supplementary Tables 1, 2).

To investigate whether the heartbeat deceleration was modulated by the trial order and auditory conditions across vigilance states, we used linear mixed-effects models with normalized RR interval (RR interval) as the dependent variable, Auditory Condition (synch, asynch, isoch), Trial Order ($RR_{om}$, $RR_{+1}$, $RR_{+2}$) and Vigilance State (AWAKE, N1, N2, N3, REM) as fixed factors and Subject as the random factor. This allowed for the comparison across vigilance states despite missing values for some participants in a given vigilance state. We first used the following model (Model 1):

$$RR\ interval \sim Auditory\ Condition * Trial\ Order + Vigilance\ State$$
$$* Auditory\ Condition + Vigilance\ State * Trial\ Order + (1|Subject)$$

The analysis revealed significant main effects of Auditory Condition ($p = 3.4 \times 10^{-11}$) and Trial Order ($p = 7.0 \times 10^{-4}$). The interaction Auditory Condition*Trial Order ($p = 1.0 \times 10^{-3}$) also significantly explained the normalized RR intervals. Conversely, the main effect of Vigilance State and

the interactions of Vigilance State*Auditory Condition and Vigilance State*Trial Order were not significant ($p > 0.05$).

We then used a triple interaction model of Auditory Condition*Trial Order*Vigilance State (Model 2):

$$RR\ interval \sim Auditory\ Condition * Trial\ Order * Vigilance\ State + (1|Subject)$$

Here, the main effects or interactions were not significant ($p > 0.05$) with the exception of Auditory Condition ($p = 8.0 \times 10^{-4}$).

Post-hoc paired Wilcoxon signed-rank tests with Bonferroni correction for multiple comparisons across conditions corroborated that in the synch condition (Fig. 2, red line), omissions elicited a long-lasting heart rate deceleration, with higher RR interval during (AWAKE: $p = 1.2 \times 10^{-5}$; N1: $p = 1.2 \times 10^{-5}$; N2: $p = 2.4 \times 10^{-5}$; N3: $p = 1.0 \times 10^{-3}$; REM: $p = 8.5 \times 10^{-4}$) and immediately after (AWAKE: $p = 1.5 \times 10^{-5}$; N1: $p = 6.1 \times 10^{-5}$; N2: $p = 2.1 \times 10^{-5}$; N3: $p = 3.0 \times 10^{-3}$; REM: $p = 6.1 \times 10^{-5}$) the omission than before the omission across all vigilance states.

These results suggest that the heartbeat deceleration upon omission was specific to the synch condition and occurred independent of the vigilance state, across wakefulness and all sleep stages.

### Neural omission response

In the EEG analysis, 23 participants were eligible for wakefulness (AWAKE) and N2 sleep as well as 12 for N1 sleep, 14 for N3 sleep, and 13 for REM sleep

(Supplementary Tables 1, 2). Since the sample size estimation for obtaining statistically significant results in the EEG comparisons (Methods. Sample size estimation) revealed a minimum sample size of 17 participants, we did not compare the EEG omission evoked responses for N1, N3 or REM sleep wherein the sample size criterion was not met.

Sounds elicited auditory evoked potentials (AEPs) in all sleep stages (Fig. 3) with no differences in the AEPs between the isoch and asynch conditions (cluster permutation statistical analysis; $p > 0.05$, two-tailed) in wakefulness and N2 sleep. We refrained from performing these comparisons in the synch condition as the AEPs in that case were contaminated with the response to the heartbeat.

To investigate the effect of cardio-audio synchronicity on omission responses, we derived HEPs during sound omissions (OHEPs) in the synch and asynch conditions for wakefulness and N2 sleep. Average OHEPs were calculated by extracting epochs from the continuous EEG recordings that were time-locked to the first R peak of the ECG signal during omissions. As an additional control condition for this analysis, we extracted a random selection of R peaks in the ECG signal to derive average HEPs from the continuous EEG recordings in the baseline condition.

In the wakefulness session, we expected to observe differences in the HEPs when comparing the synch to asynch and the synch to baseline condition, as a consequence of the predictability of sound onset in the synch condition, based on the fixed delay between R peaks and sounds[30]. For the synch vs asynch comparison (Fig. 4a), the cluster-based permutation test ($p < 0.05$, two-tailed) revealed a significant negative cluster ($p = 0.027$, Cohen's $d = 0.841$) at 158 ms to 216 ms following R peak on anterior-central scalp electrodes. Similar results were observed in the synch vs baseline comparison (Supplementary Fig. 1a) while the asynch vs baseline comparison showed no significant differences (Supplementary Fig. 1b).

In N2 sleep, we similarly identified a negative cluster upon comparing the OHEPs for the synch and asynch conditions (Fig. 4b) at 332 ms to 500 ms ($p = 0.044$, Cohen's $d = 0.762$) following R peak onset. In N2 sleep, we did not find significant results neither in the comparison of synch vs baseline nor in the contrast of asynch vs baseline.

Overall, our results indicate that the fixed heartbeat-to-sound interval induced auditory prediction observed as a neural surprise MMN-like response upon regularity violation during omissions in both wakefulness and N2 sleep.

As a further validation of the existence of an auditory predictive processing mechanism during wakefulness and sleep, we tested whether fixed sound-to-sound intervals induced an expectation of upcoming auditory stimuli, violated upon omission. To do so, we derived the responses during sound omissions in the isoch and asynch conditions in wakefulness and N2 sleep. Average sound-based omission evoked potentials (OEPs) were calculated by extracting epochs from the continuous EEG recordings that

### Table 1 | Group averaged sleep characteristics

| N = 25 | Mean | SD |
|---|---|---|
| Total Bed Time (min) | 420.3 | 93.8 |
| Total Sleep Time (min) | 302.4 | 107.9 |
| Sleep Onset (min) | 24.6 | 19.1 |
| Sleep Efficiency (%) | 69.8 | 12.9 |
| N1 (%) | 12.9 | 4.5 |
| N2 (%) | 57.4 | 6.9 |
| N3 (%) | 15.5 | 8.8 |
| REM (%) | 14.1 | 5.5 |

Total Bed Time calculated based on Lights Off and Lights On time points. Total Sleep Time computed as the period between the Sleep Onset and final awakening, uncovered by the sleep staging, excluding wakefulness periods but including possible micro-arousal periods during the night. Percentages of sleep stage periods (N1, N2, N3, REM) during the session were computed in relation to the Total Sleep Time. Sleep Efficiency (%) = Total Sleep Time/Total Bed Time for each participant × 100.
SD Standard Deviation.

**Fig. 2 | Cardiac omission response in wakefulness and sleep.** Singe-subject (top panel) and grand average (bottom panel) normalized RR intervals prior ($RR_{-1}$), during ($RR_{om}$), and following ($RR_{+1}$ and $RR_{+2}$) sound omissions across auditory conditions (synch: red line; asynch: black line; isoch: blue line) in the wakefulness session (AWAKE) and all sleep stages (N1, N2, N3, REM). Error bars indicate the half SEM. Linear mixed models and post-hoc Wilcoxon signed-rank tests highlighted a heart rate deceleration ($p < 0.05$) upon sound omission in the synch condition across all vigilance states.

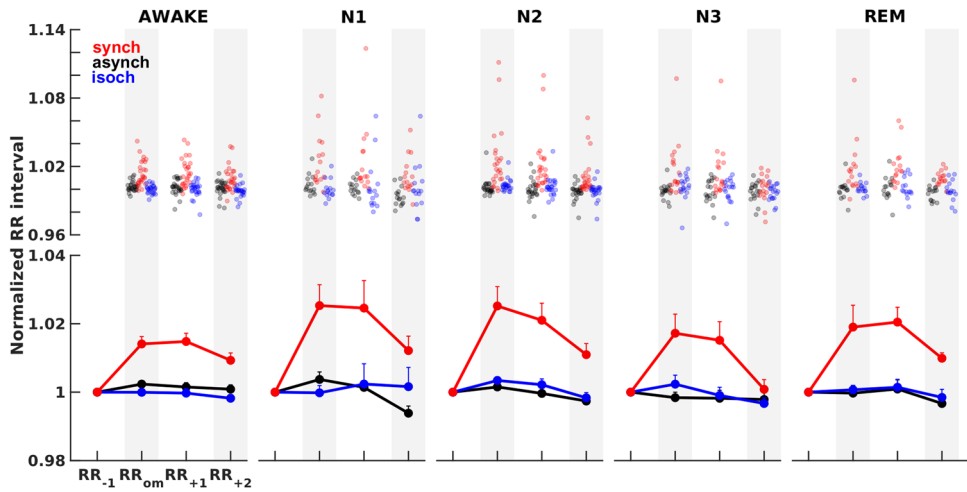

**Fig. 3 | Auditory evoked potentials (AEPs) in wakefulness and sleep.** Grand average AEPs in the isoch condition on EEG electrodes located along the midline in wakefulness (AWAKE) and all stages of sleep (N1, N2, N3, REM). AEPs in wakefulness display the expected electro-physiological signatures such as the N100 as the first and highest in amplitude component and the expected shift to later P200 and N550 components across sleep stages. Asynch condition AEPs are not depicted due to the high similarity to the isoch condition and synch condition AEPs are not depicted as they would be superimposed with the heartbeat evoked potential.

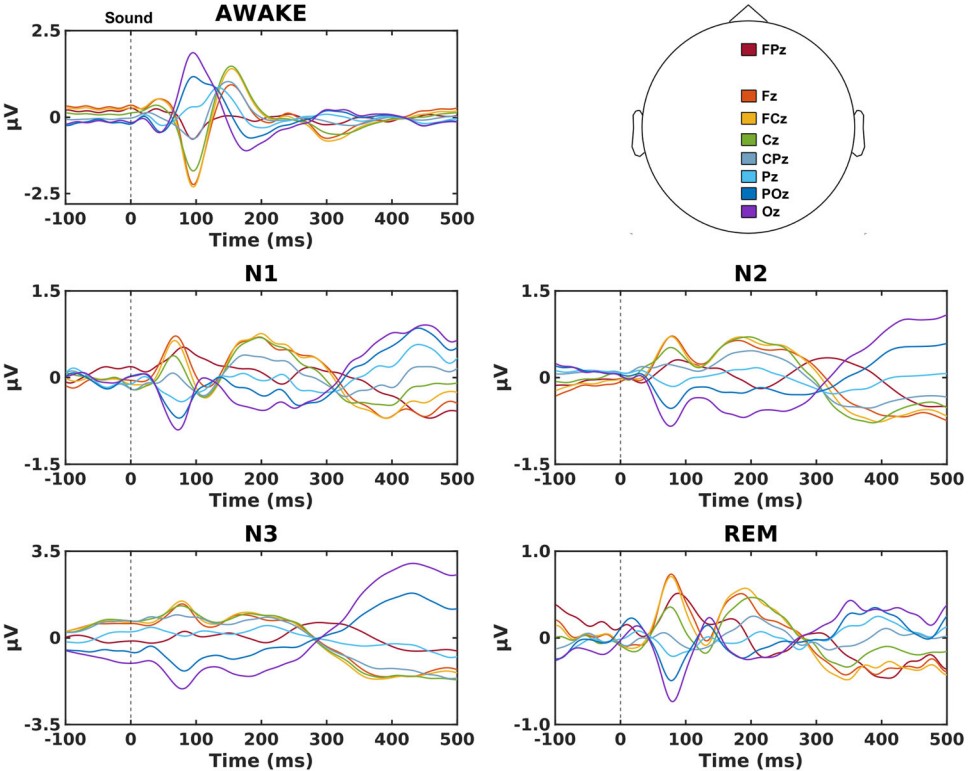

were time-locked to the average SS interval. A random selection of epochs were extracted from continuous baseline recordings, such that the latencies between epoch onset and closest heartbeat (i.e., R peak) were matched to the trial onsets in the sound-based isoch and asynch conditions at the single-trial level.

Based on previous reports of neural responses to omissions within regular auditory sequences during wakefulness (e.g.,[12]), here, we expected central negativity in the isoch condition (where temporal regularity existed in sound stimuli) between 100 ms and 250 ms. In the wakefulness session, the cluster-based permutation test ($p < 0.05$, two-tailed) comparing the OEPs in the isoch $vs$ asynch condition yielded a significant negative cluster ($p = 0.043$, Cohen's $d = 1.011$) at 115 ms to 160 ms following expected sound onset in anterior-central scalp electrodes (Fig. 4c). The existence of an omission response in the isoch condition was further confirmed by the isoch $vs$ baseline comparison (Supplementary Fig. 1c). Despite the absence of regularity in the asynch condition, the asynch $vs$ baseline comparison in wakefulness revealed a significant negative cluster approximately at the expected sound onset (~0 ms; Supplementary Fig. 1d). This last result indicates some degree of prediction of upcoming sounds in the asynch sequence during wakefulness despite the absence of temporal expectation, plausibly due to the pseudo-regularity of the auditory sequence.

In N2 sleep, the isoch $vs$ asynch OEPs comparison revealed significant differences that were largely similar to wakefulness, at least in terms of latency relative to expected sound onset. In more detail, statistical evaluation of the isoch and asynch condition differences (Fig. 4d) identified a significant negative cluster ($p = 0.046$, Cohen's $d = 0.651$) at 85 ms to 226 ms following expected sound onset localized to posterior scalp electrodes. Similar to wakefulness, the isoch $vs$ baseline comparison in N2 sleep further confirmed the surprise response upon violation of the isoch auditory regularity (Supplementary Fig. 2a). Unlike wakefulness, the asynch $vs$ baseline comparison revealed no statistically significant differences in N2 sleep, pointing to the lack of omission responses in the absence of temporal regularity between auditory stimuli (Supplementary Fig. 2b).

Our results in wakefulness and N2 sleep suggest that in the isoch condition, the fixed SS interval induced an expectation of upcoming sounds and resulted in a neural surprise response upon violation of the regularity rule during omissions.

## Slow oscillations analysis

Since sound presentations are known to alter the background oscillatory activity in sleep, notably the SOs (0.5–1.2 Hz) during NREM sleep[40,42,43], we investigated whether the three auditory conditions had differential effects on the ongoing SOs during N2 sleep. After extracting the SOs during N2 sleep for each experimental condition at electrode Cz (Fig. 5a), we identified the closest positive peak latency of each SO to a given sound or R peak (Fig. 5b) and computed the median sound-to-SO latency for all auditory conditions and median R peak-to-SO latency for all auditory conditions and the baseline (Fig. 5c) for latencies between −800 ms and 800 ms[33].

Statistical assessment using an 1 × 3 repeated measures Friedman test on the mean sound-to-SO latencies yielded significant differences across auditory conditions (Fig. 5c; $\chi^2_{(1,3)} = 7.9$, $p = 0.019$). Post-hoc Wilcoxon signed-rank tests confirmed lower latencies in synch compared to asynch ($p = 0.009$) and in isoch compared to asynch ($p = 0.026$). These results suggested a possible readjustment of the SOs with respect to sound onset depending on the regularity condition (Fig. 5c) since when regularity was present, either in the synch or isoch condition, SOs tended to align to the sound onset. Because of the fixed RS delay in the synch condition, the alignment between sound onset and SOs was also reflected in a lower R peak to SO peak in the synch compared to the asynch and isoch conditions (Fig. 5c). In order to rule out that this SO readjustment was due to a specific relation between R peak and SO irrespective of sound presentation, we carried out a 1 × 4 repeated measures Friedman test on the mean R peak-to-SO latencies which revealed no significant differences across conditions (Fig. 5c; $\chi^2_{(1,4)} = 7.4$, $p = 0.062$), suggesting that potential readjustment of SOs was specific to auditory regularities and not to cardiac input.

**Fig. 4 | Neural omission responses in wakefulness and N2 sleep.** OHEPs (time-locked to the R peak) and OEPs (time-locked to mean SS interval) comparisons during sound omissions for the synch (red lines) and isoch (blue lines) conditions compared to the asynch condition (black lines) during wakefulness (AWAKE) and N2 sleep (N2). Top panels show grand average (N = 23) EEG waveforms averaged over all electrodes in the significant negative clusters. Shaded regions indicate ±SEM across participants. Middle panels display the number of electrodes within each significant cluster derived from cluster permutation statistical analysis (p < 0.05, two-tailed). Bottom panels demonstrate topography differences at negative cluster peaks with significant electrodes highlighted in black. Significant differences are observed between the synch and asynch OHEPs in wakefulness (**a**) and N2 sleep (**b**), suggesting that the synch cardio-audio regularity induces a modulation of OHEPs both in wakefulness and N2 sleep. Significant differences between the isoch and asynch condition OEPs in wakefulness (**c**) and N2 sleep (**d**) demonstrate that the isoch regularity produces an expectation of incoming sounds observed as a MMN response at ~150 ms, in wakefulness and N2 sleep.

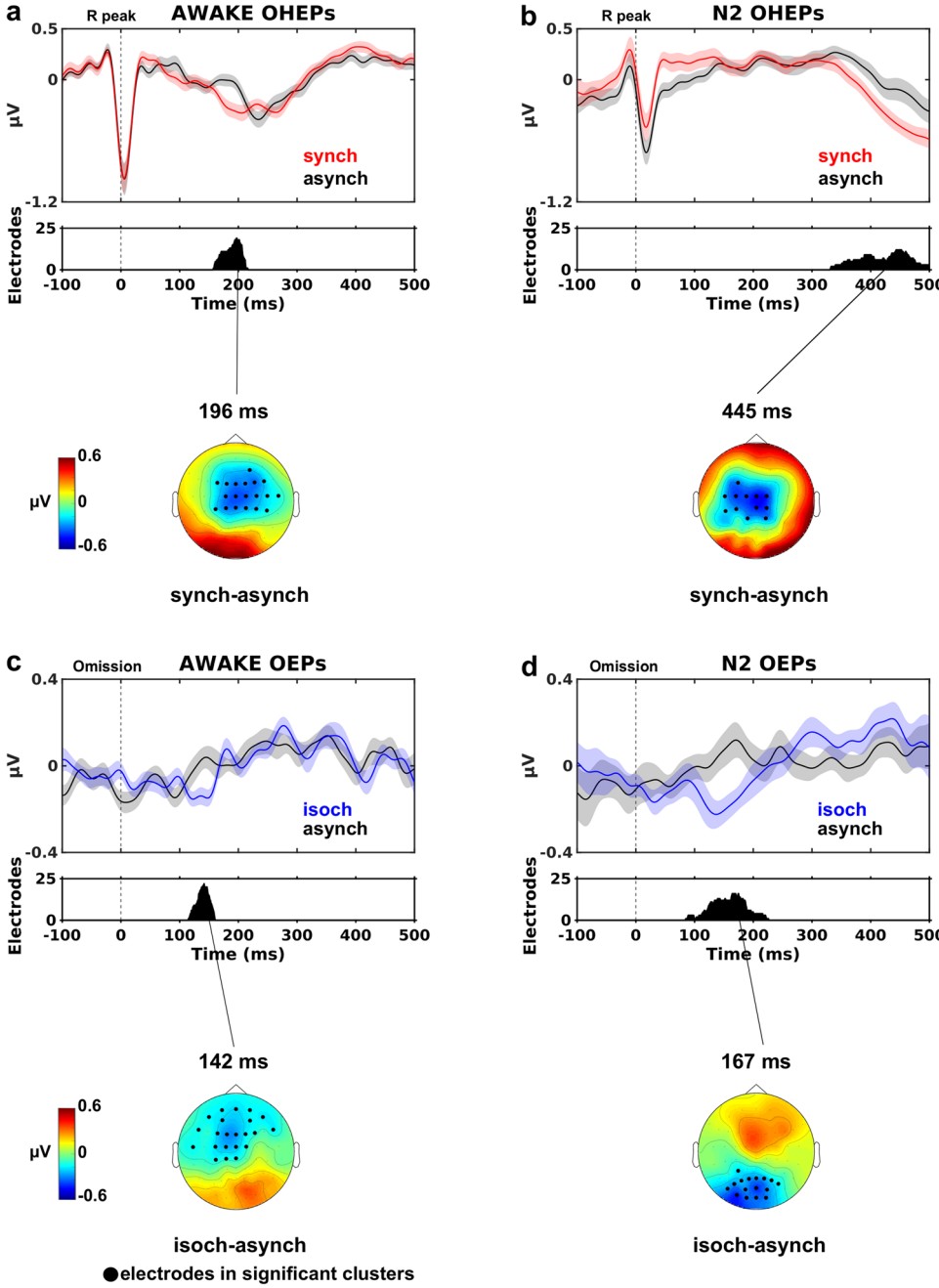

## Quality control analyses

We conducted a series of control analyses on the relation between sound onsets, heartbeat and sound onsets and the heart rate across auditory conditions in order to identify possible factors influencing the cardiac and neural response to sound omissions. Here, unless otherwise specified, we report the results in the cohort of participants included in the ECG analysis, the results of the EEG cohort for wakefulness and N2 sleep being very similar.

In the synch condition, the average RS interval was 52.3 ms (SEM = 0.1 ms) for sound trials, and −2.6 ms (SEM = 2.6 ms) for the isoch and asynch conditions across wakefulness and all sleep stages (Supplementary Fig. 3a). We observed the expected lower RS interval variability in the synch relative to the two other conditions (isoch and asynch) with values of 0.2 ms (SEM = 0.0 ms) for the synch condition and 10.6 ms (SEM = 0.7 ms) for the isoch and asynch conditions across wakefulness and all sleep stages. This was confirmed by one way repeated

measures Friedman tests with factor Auditory Condition (synch, asynch, isoch) (Supplementary Fig. 3b; AWAKE: $\chi^2_{(1,3)} = 39.1$, $p = 3.3 \times 10^{-9}$; N1: $\chi^2_{(1,3)} = 22.5$, $p = 1.3 \times 10^{-5}$; N2: $\chi^2_{(1,3)} = 36.0$, $p = 1.5 \times 10^{-8}$; N3: $\chi^2_{(1,3)} = 27.0$, $p = 1.4 \times 10^{-6}$; REM: $\chi^2_{(1,3)} = 22.5$, $p = 1.3 \times 10^{-5}$; all post-hoc paired Wilcoxon signed-rank tests showed lower variability in the synch compared to the asynch or isoch condition with $p < 0.0005$). In addition, as expected, the mean SR interval within omission was more variable in the asynch and isoch conditions compared to the synch condition (Supplementary Fig. 3d; AWAKE: $\chi^2_{(1,3)} = 39.3$, $p = 2.9 \times 10^{-9}$; N1: $\chi^2_{(1,3)} = 22.5$, $p = 1.3 \times 10^{-5}$; N2: $\chi^2_{(1,3)} = 36.3$, $p = 1.3 \times 10^{-8}$; N3: $\chi^2_{(1,3)} = 27.1$, $p = 1.3 \times 10^{-6}$; REM: $\chi^2_{(1,3)} = 22.8$, $p = 1.1 \times 10^{-5}$; post-hoc Wilcoxon signed-ranked tests confirmed significant differences with $p < 0.0005$). This first series of control analyses demonstrated that the experimental manipulation, inducing an online fixed temporal alignment between R peak and sound in the synch and a variable one in isoch and asynch, was successful.

### a Slow oscillation extraction

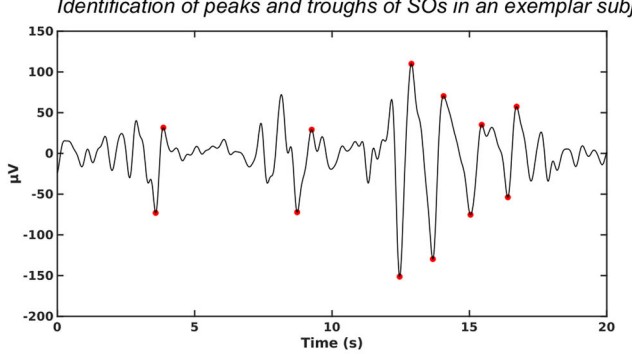
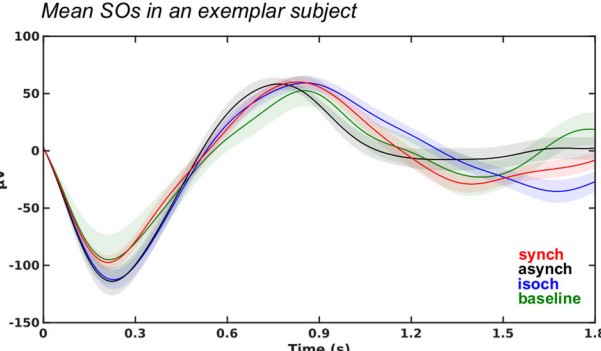

### b Slow oscillation selection and realignment

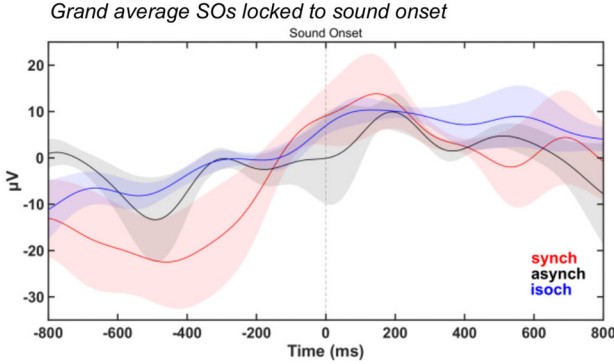
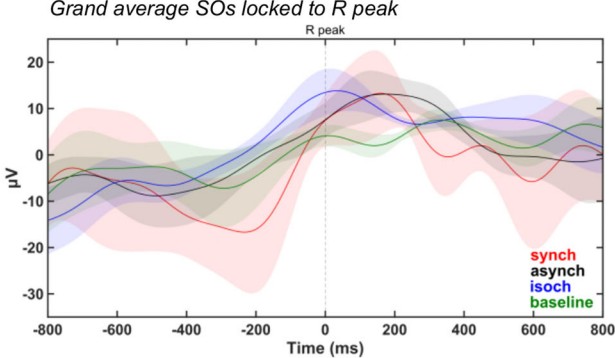

### c Slow oscillation peak to sound and R peak latency statistical analysis

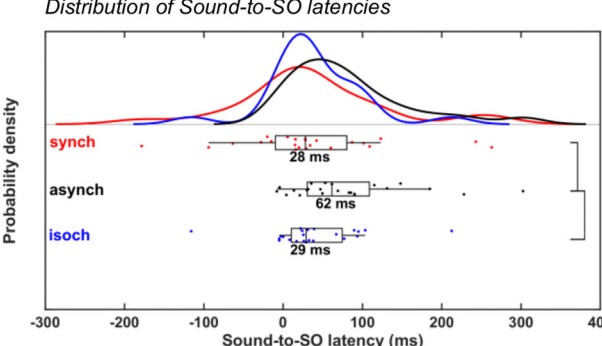
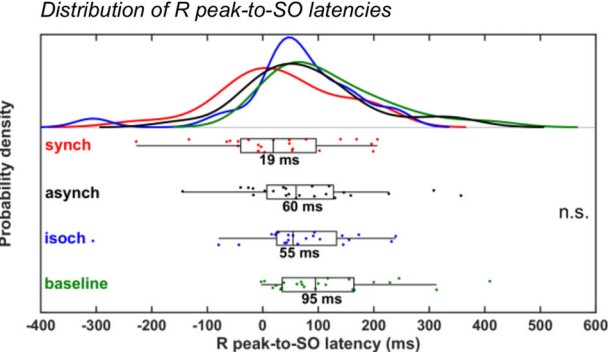

**Fig. 5 | Modulation of slow oscillation (SO) activity by auditory stimulation in N2 sleep. a** Extraction of SOs from the continuous N2 sleep data and the corresponding mean SO in all experimental conditions at electrode Cz for one exemplar subject. **b** Grand average ($N = 23$) SO waveforms at electrode Cz centered around sound onset and R peak in the synch, asynch, isoch, and baseline conditions during N2 sleep. **c** Distribution and single subject sound-to-SO median latency and R peak-to-SO median latency with statistically significant differences ($p < 0.05$; black vertical lines) across conditions. SO peaks are more likely to occur closer to the sound onset in the synch and isoch conditions than in the asynch conditions. These results are not trivially driven by a temporal proximity between R peaks and SO peaks in the synch condition. n.s. = not significant.

The analysis of auditory regularity based on the SS interval confirmed higher variability during the synch and asynch compared to the isoch condition (Supplementary Fig. 3f; AWAKE: $\chi^2_{(1,3)} = 39.3$, $p = 2.9 \times 10^{-9}$; N1: $\chi^2_{(1,3)} = 22.5$, $p = 1.3 \times 10^{-5}$; N2: $\chi^2_{(1,3)} = 37.3$, $p = 7.8 \times 10^{-9}$; N3: $\chi^2_{(1,3)} = 19.4$, $p = 6.0 \times 10^{-5}$; REM: $\chi^2_{(1,3)} = 17.7$, $p = 1.0 \times 10^{-4}$; corroborated by post-hoc Wilcoxon signed-rank tests with $p < 0.0005$) with an average variability value of 0.9 ms in wakefulness and 3.4 ms across sleep stages for the isoch condition (Supplementary Fig. 3e). This second analysis confirmed that the isoch condition was characterized by highly regular sound-to-sound intervals in comparison to the other conditions.

The heartbeat did not change across experimental conditions during N2, N3, or REM sleep as shown by $1 \times 4$ repeated measures Friedman tests on average RR intervals with factor Condition (synch, asynch, isoch, baseline) (Supplementary Fig. 3g; $p > 0.05$). Conversely, the same analysis performed in wakefulness and N1 sleep revealed significant differences in RR intervals (Supplementary Fig. 3g; AWAKE: $\chi^2_{(1,3)} = 17.4$, $p = 6.0 \times 10^{-4}$; N1: $\chi^2_{(1,3)} = 17.0$, $p = 7.0 \times 10^{-4}$). Post-hoc Wilcoxon signed-rank tests uncovered that significantly reduced RR intervals were specific to the baseline condition (AWAKE: $p = 0.002$ for synch vs baseline, $p = 0.001$ for asynch/isoch vs baseline; N1: $p = 0.013$ for synch vs baseline, $p = 0.008$ for asynch/isoch vs baseline) while no differences were observed across auditory conditions ($p > 0.05$).

Finally, for the EEG cohort, we carried out a control analysis on the ECG waveforms to exclude the potential confound of a different degree of contamination of the ECG artifacts in the EEG between conditions of interest upon interpreting the differential EEG signals locked to the R

peak. To this aim, we performed a time-wise ECG waveform comparison of the grand averaged ECG trials time-locked to R peaks during sound omissions (Supplementary Fig. 4). Statistical analysis based on non-parametric cluster permutation statistics ($p < 0.05$, two tailed) contrasting paired experimental conditions (synch, asynch, isoch, baseline) revealed no significant differences in wakefulness and N2 sleep for any of the comparisons.

Overall, these last series of analyses suggest that the heartbeat and ECG characteristics were well matched across auditory conditions and as a consequence, they should not confound the cardiac and neural differential responses across regularity types.

## Discussion

We investigated the neural and cardiac correlates of cardio-audio regularity encoding in wakefulness and sleep by administering sounds in synchrony to the ongoing heartbeat (synch), at fixed temporal pace (isoch), and in a control condition without specific temporal regularity (asynch) while maintaining matched average sound-to-sound intervals across conditions. We tested whether auditory regularity encoding would result in a prediction error signal as measured by the cardiac and neural signals upon unexpected omitted sounds. A strong cardiac deceleration after sound omission during the cardio-audio sequence suggested an enhanced modulation of the cardiac activity upon unexpected omissions relative to the other auditory regularity types in all vigilance states. Similarly, neural responses to sound omissions revealed that auditory regularities induced prediction of upcoming sounds both when sounds occurred at a fixed pace and when temporally synchronized to the ongoing heartbeat during wakefulness and N2 sleep. Analysis of the SOs during N2 sleep revealed a reorganization of the ongoing background brain activity both when sounds occurred in synchrony with the ongoing heartbeat and at a fixed temporal pace.

We observed a cardiac deceleration upon omission across vigilance states (Fig. 2), specific to the synch condition. This result is reminiscent of an attention reorientation response following an unexpected and potentially dangerous event, a parasympathetically-driven effect often reported in conditioning paradigms[44–46]. Heart rate deceleration has also been reported in the context of a startle reaction and a physiological freezing response, often accompanied by pupil dilation and skin conductance alterations[47–51]. In N2 sleep, this cardiac activity modulation could be related to cholinergic system engagement. The cholinergic system is known to modulate arousal, also observed in rats when awakened from NREM sleep by activation of basal forebrain cholinergic neurons using chemical and optogenetic techniques[52,53]. Of note, we found similar deceleration across vigilance states although acetylcholine levels are reduced in N2 sleep and increased in REM sleep compared to wakefulness[54,55].

The heart rate deceleration in the synch condition could also be related to a top-down adjustment of cardiac rhythm in order to account for the unexpected silence. As a sound is predicted following a heartbeat (in the synch), the omission may prolong the generation of the next heartbeat within physiologically plausible boundaries so as to 'wait' for a delayed auditory stimulus within the sequence, followed by a rapid readjustment to the original rhythm upon subsequent sound presentations. In a previous study investigating the cardiac response to sound omission as a function of heartbeat-to-sound onset delay and of interoceptive vs exteroceptive attention, a similar cardiac deceleration was reported only in the condition of external attention[29]. While this result seems at odds with our finding of preserved cardiac deceleration across vigilance states (and potentially attentional resources), a straightforward comparison is prevented by our lack of control of the focus of attentional resources during wake, also shown to be preserved in sleep[56,57].

The neural response to sound omissions was observed as a negative difference in fronto-central electrodes between the synch and asynch conditions at 158 ms to 216 ms (Fig. 4a) and the isoch and asynch conditions at 115 ms to 160 ms (Fig. 4c). These latencies are well-matched to classic MMN responses observed in wakefulness between 100 and 200 ms post-stimulus onset[2,10], upon consideration of the heartbeat to sound latency of

52 ms in the synch condition. Of relevance, the latencies observed here parallel findings from our previous investigation of the synch vs asynch comparison where significant differences were observed in the synch vs asynch comparison at overlapping latencies between 153 and 278 ms after R peak onset[30]. Our results also resemble previous omission responses peaking at ~170 ms and arising from the comparison of omission responses with different level of predictability within auditory sequences[12]. Other reports on the neural correlates of predictive processing in auditory regularities have explored different aspects of the omission response. SanMiguel and colleagues investigated the differential responses to button presses that resulted in sounds or omissions and showed similar N100 responses and cortical sources to unpredictable sounds vs unpredictable omissions[15], an observation replicated by other groups[58,59]. Finally, while we used largely similar pre-processing (i.e. filters) and experiment implementation (i.e., online and offline reference) as in Chennu et al.[12], this was not the case for other studies with which direct comparisons are unwarranted.

In our study, the differences between omission responses and baseline also included earlier components in fronto-central electrodes: for the synch condition starting at 57 ms (Supplementary Fig. 2a) and for the isoch condition starting at 52 ms (Supplementary Fig. 2c). We interpret these early differences in light of previous reports suggesting the formation of a sensory template at the predicted sound onset occurring in the same period as the auditory N100 response[60] and localized in the auditory cortex[15,16]. The fact that these early differences did not occur when comparing omissions between auditory regularities (Fig. 4) is likely due to the type of generated prediction: upon omissions during auditory regularities, the expectation of a sound is always violated ('what' is expected) and therefore the prediction error signal at early latencies (~50 ms; Supplementary Fig. 1a, c) is likely canceled out. The contrast between omissions during auditory regularities (Fig. 4) instead uncovers the prediction error signal of 'when' a sound is expected which is generated during the isoch and synch conditions. This line of reasoning on the occurrence of an auditory prediction (prediction of 'what') in all auditory sequences is also confirmed by the differential response between the omission responses in the asynch vs baseline (Supplementary Fig. 1d).

During N2 sleep, we identified long-lasting omission responses for conditions of auditory regularities (Fig. 4d; isoch vs asynch comparison) at 85 ms to 226 ms after expected sound onset and in cardio-audio regularities (Fig. 4b; synch vs asynch comparison) at 322 ms to 500 ms after R peak onset. The neural responses during the isoch vs asynch condition revealed a posterior negative polarity difference at latencies closely matching our results in wakefulness, and, to the best of our knowledge, provide the first account of omission responses in sleep. Previous reports using deviant sounds instead of omissions embedded within regular auditory sequences, demonstrated MMN responses in NREM sleep at similar latencies[6,61–63]. Importantly, similar to these previous MMN studies in sleep, our omission responses in N2 sleep are characterized by a positive difference at fronto-central electrodes[6,61–63].

The observation of differential responses in the synch vs asynch OHEPs occurring at much later latencies (Fig. 4b; starting at ~300 ms) compared to wakefulness (Fig. 4a; starting at ~150 ms) despite similar fronto-central negative polarity could be intrepreted in light of the difference in average RR intervals in wakefulness and N2 sleep (Supplementary Fig. 3g; approximately 800 ms and 1000 ms, respectively). As a result of this slower heart rate in N2 sleep in comparison to wakefulness, the auditory stimuli in the synch condition, occurring at 52 ms after the R peak, fell within an earlier systole phase in sleep than in wakefulness. Accumulating evidence using a variety of stimulus modalities suggests that the precise timing at which sensory stimuli are administered during the cardiac phase impacts the processing of such stimuli, possibly due to the interference with baroreceptor firing in the systolic but not the diastolic phase of the cardiac cycle[19,64–66] and to differences in cortical excitability[20]. Later phases of the systole period in wakefulness compared to N2 sleep may be facilitatory for sound and omission processing, which would plausibly result in earlier differential responses in the synch vs asynch OHEPs in wakefulness (Fig. 4a) vs N2 sleep (Fig. 4b). This line of reasoning can also offer a possible

explanation for the later latencies at which we observe differential responses in the synch vs asynch OHEPs in N2 sleep (Fig. 4b) compared to the isoch vs asynch OEPs in N2 sleep (Fig. 4d).

The well-known modulation of background oscillatory activity by auditory stimulation in NREM sleep prompted us to look at the impact of auditory regularity processing on SO activity (0.5–1.2 Hz oscillations[42,67,68]) in N2 sleep. Here, this influence was demonstrated by a significantly reduced median latency between the sound stimulus onset and the peak of SOs for the synch and isoch conditions compared to the asynch condition (Fig. 5c). This indicates that sound presentations induced a modulation of slow oscillatory activity in N2 sleep in our subjects in such a way that, when auditory prediction could be generated (i.e. synch and isoch), sound onset was more likely to occur close to the SO peaks. This evidence is reminiscent of the closed-loop auditory stimulation literature wherein temporary synchronization of sound stimuli to ongoing SOs induced an enhancement of the SO rhythm during NREM sleep[40,42,43,69]. In the present paradigm as well as in closed-loop auditory stimulation studies, the temporal proximity between sound onset and the SO positive peak suggests the existence of a preferential time window of stimulus processing which may coincide with the positive phase of the SO cycle when neuronal firing is maximal[68,70,71]. On this basis, auditory regularity encoding would induce a reorganization of the ongoing SO activity, in order to facilitate the neural processing of expected sounds in a sequence when sound onset can be predicted (see also[72] for consistent findings in associative learning bound to the SO peaks).

Of note, not only sounds have been shown to have an impact on the latency of SOs in NREM sleep but also R peaks tend to occur close to the positive peak of the SO compared to other latencies[33,34]. With this in mind, we additionally investigated the potential impact of heartbeat signals on SOs (Fig. 5b, c). In this study, we revealed no significant differences between R peak onset and SO peak across conditions. However, it should be noted that although not significant, we observed a trend of lower R peak to SO peak latencies during cardio-audio regularity compared to the other auditory conditions (Fig. 5c), possibly driven by the fixed relationship between heartbeat and sound in the synch condition. Overall, these findings suggest that SO latency modulation was specific to auditory regularities and not driven by a systematic temporal relationship between the SO peaks and the ongoing heartbeat.

The selection of a ~ 50 ms delay between the detected R peaks and administered sounds led by construction to the investigation of auditory regularity encoding within the systole period in the synch condition. The observed results might be related to this specific temporal period and not necessarily generalize to other latencies of sound administration after the R peak, particularly during the diastole period[19,64,73]. Future studies will focus on the cardiac and neural correlates of the cardio-audio coupling along different phases of the cardiac cycles. In addition, the current results do not allow for a strict identification of the latency at which the cardiac and neural signals are integrated at the neural level. Indeed, by imposing a fixed R peak-to-sound interval in the synch condition, we indirectly impose a stable temporal relationship between any point within the heartbeat cycle - due to its relatively fixed periodicity - and the sound onset. Considering this periodicity, we cannot exclude that the ECG and EEG omission responses may arise from cardiac related information conveyed to the brain at earlier latencies than the R peak. Another possible limitation relates to the experimental protocol for sleep data acquisition. While participant selection involved an interview assessing each individual's sleep quality, we did not evaluate our cohort's general sleep health in a more systematic way using sleep quality assessment questionnaires or actigraphy monitoring, as recommended in sleep research[74]. This, along with the auditory stimulation performed and the absence of an adaptation night sleep, could have resulted in a participant cohort with variable sleep quality[75] and have given rise to the low sleep efficiency (69.8%) compared to typically >80% sleep efficiency in healthy unperturbed sleep (e.g.[36,76]). In future work, we will improve participant sleep assessment and sleep quality by including an adaptation night[74] before the data acquisition nights which we speculate will yield improved EEG data availability and will likely result in neural omission responses in

the remaining sleep stages (N1, N3, REM), as was observed in the ECG-based cardiac omission response.

To summarize, we first studied the role of heartbeat signals in auditory regularity processing focusing on sound omissions, which enabled the investigation of cardio-audio integration free from bottom-up auditory stimulus contributions, differently to previous studies in sleep employing deviant sounds in MMN investigations (e.g.[4–6]). Second, the proposed experimental paradigm enabled the estimation of the cardio-audio omission response while matching for possible cardiac related artifacts which were present in all experimental conditions in the within-subject comparison of the neural response locked to the R peak of the ECG signal. Third, by investigating both wakefulness and sleep in the same healthy volunteers, we now demonstrate that the human brain infers on the temporal relationship across cardiac and auditory inputs in making predictions about upcoming auditory events across vigilance states. Fourth, we identified a cardiac deceleration as a result of violation detection occurring in parallel to the neural violation response during sound omission. This omission response during N2 sleep was also accompanied by SO reorganization, representing a possible mechanism through which the brain aligns periods of high neuronal excitability to the expected sound onset. The cardio-audio synchronicity created ad hoc in the experimental environment might reflect a real life readjustment of the heartbeat rhythm in order to optimize the temporal relationship between bodily signals and exteroceptive inputs for optimal sensory encoding across vigilance states. These results complement recent accumulating evidence of cardiac signal based markers for assessing the degree of preserved cognitive functioning across a variety of disorders of consciousness[41,77].

Collectively, the present results suggest that the human brain can keep track of temporal regularities between exteroceptive inputs, and across interoceptive and exteroceptive inputs during both wakefulness and sleep. Our findings support theories of an interoceptive predictive coding mechanism[21,22,78]. To the best of our knowledge, this is the first study to investigate auditory regularity processing using omissions and to offer evidence for a potential role of interoceptive inputs under a predictive coding framework in sleep. The conscious and unconscious brain may implicitly process relationships across interoceptive and exteroceptive inputs in order to optimize the signaling and prediction of potential upcoming dangers.

## Materials and methods
### Ethics statement
All ethical regulations relevant to human research participants were followed. Approval for the study (Project-ID: 2020-02373) was obtained by the local ethics committee (La Commission Cantonale d'Ethique de la Recherche sur l'Etre Humain), in accordance with the Helsinki declaration.

### Sample size estimation
The number of participants selected for this study was based on the results of our previous study in wakefulness[30]. Sample size was derived from the electrodes within significant clusters and the latencies at which they were observed when comparing OHEPs in the synch vs asynch comparison. After fixing the probability threshold for rejecting the null hypothesis to 0.05 (two-tailed), we simulated 5000 random replications of the neural responses to sound omissions. These simulations were based on extracting random samples from a multivariate normal distribution with mean and variance estimated from previous data (in accordance with https://osf.io/rmqhc, documented in the publically available Fieldtrip toolbox, https://www.fieldtriptoolbox.org/example/samplesize/). Cluster-based permutation statistical analysis on each of the simulated comparisons, provided a power >0.90 (i.e., the percentage of times the results were significant) with a sample size higher than seventeen for the healthy cohort ($N > 17$). Here, we decided to recruit twenty-six volunteers, in order to account for possible participant exclusion due to a higher likelihood of equipment malfunction, excessive artifactual trials or channels during full-night sleep recordings.

## Human participants

Twenty-six self-reported good sleeper volunteers took part in both the wakefulness and sleep arms of this study. Participants were considered eligible if they had no history of psychiatric, neurological, respiratory or cardiovascular conditions, no sleep apnoea, and a regular sleep schedule, evaluated during a phone interview. Hearing conditions were an additional exclusion criterion. All participants gave written informed consent and received approximately 150 Swiss Francs as monetary compensation.

## Experimental design

A two-way crossover experimental design was implemented in this study. Participants attended one wakefulness and one sleep session on two occasions separated by a minimum of one day and a maximum of ten days (wakefulness session first for 12 out of 26 participants). In both sessions, participants were instructed to passively listen to the administered sound sequences. They were naïve to the experimental manipulation, as suggested by informal verbal inquiry regarding the experimental design after the experiment. At the end of the second session, and if desired, participants were debriefed about the purpose of the experiment.

The sleep session recordings took place in a sound-attenuated hospital room equipped with a comfortable hospital bed to allow for overnight sleep recordings. During the sleep session, participants arrived at the laboratory at approximately 9 pm and following set-up preparation, they were instructed to lie down and inform the experimenters when they were ready to sleep. Lights were then switched off, the auditory stimulus administration commenced and the volunteers were left alone to naturally fall asleep. Although participants had the liberty to leave at any time, we explained that their inclusion in the study required a minimum of four hours of continuous data acquisition after sleep onset. They were free to choose to spend the night at the sleep laboratory and to be woken up at a desired time or by approximately 7 am the next morning, at which time lights were switched on.

In both sessions, participants were equipped with electrodes for heartbeat (ECG), eye movement (EOG), and EEG recordings (see below). For the sleep session, additional electrodes for submental electromyography (EMG) were attached, in accordance with the 2007 AASM guidelines for sleep scoring[79]. In-ear phones (Shure SE 215, Niles, IL) were utilized during both sessions instead of external headphones, in order to increase sound attenuation, subject comfort during sleep and to prevent physical contact with and thus displacement of EEG cap and electrodes. The online EEG, ECG, EOG and EMG were continuously monitored by the experimenters to ensure effective stimulus administration, data acquisition, heartbeat detection, and sleep quality throughout both sessions.

## Stimuli

Sound stimuli were 1000 Hz sinusoidal tones of 100 ms duration (including 7 ms rise and fall times) and 0 μs inter-aural time difference. A 10 ms linear amplitude envelope was applied at stimulus onset and offset to avoid clicks. Stimuli were 16-bit stereo sounds sampled at 44.1 kHz and were presented binaurally with individually adjusted intensity to a comfortable level for wakefulness. A considerably lower than wakefulness intensity of approximately 45 dB was chosen for sleep, in order to facilitate a non-fragmented sleep session without multiple awakenings.

## Experimental procedure

During the wakefulness session, volunteers sat comfortably on a chair in a sound-attenuated experimental room and were instructed to keep their eyes open, avoid excessive eye blinking, body and jaw movements; the aforementioned measures served in ensuring high signal quality. Each participant was presented with four types of stimulation conditions administered in separate experimental blocks in a pseudo-random order and was asked to passively listen to the sounds while keeping the eyes fixed on a cross centrally located in the visual field. The conditions were a baseline without auditory stimulation and three auditory conditions, namely synch, asynch and isoch. During wakefulness, the baseline lasted ten minutes and was acquired prior to auditory stimulation. During sleep, numerous two-minute baseline blocks

were acquired in alternation to the sound sequences in an attempt to ensure that baseline background activity was comparable to the preceding sound stimulation during all stages of sleep. The three auditory conditions lasted five minutes each and corresponded to separate experimental blocks, which were repeated six times during wakefulness in a semi-randomized order. During the sleep session, sounds were administered for the entire length of the sleep recording in sequences of three auditory blocks always followed by a baseline (e.g. isoch-synch-asynch-baseline or synch-asynch-isoch-baseline).

## Auditory conditions

All auditory conditions consisted of the sequential presentation of 250 stimuli (80% sounds and 20% omissions) administered in a pseudo-random order wherein at least one sound stimulus intervened between two subsequent omissions. Details for each auditory condition are given below and a thorough post-hoc evaluation of the experimental manipulation is provided in the Results and Supplementary Information (Results. Quality control analyses & Supplementary Fig. 3).

In the synch condition, the temporal onset of each sound stimulus was triggered by the online detection of R peaks from raw ECG recordings. To enable effective online R peak detection, raw ECG recordings were analyzed in real-time using a custom MATLAB Simulink script (R2019b, The MathWorks, Natick, MA). The variance over the preceding 50 ms time window was computed and an R peak was detected when the online ECG value exceeded an individually adjusted 10–15 mV$^2$ variance threshold, which in turn triggered the presentation of a sound stimulus or an omission. This procedure resulted in a fixed R peak-to-sound average delay (RS interval) of 52 ms (SD = 5 ms) for wakefulness and sleep across participants, the minimum fixed delay offered by the utilized equipment in order to best approximate the condition of co-occurrence between the heartbeat and auditory stimuli over time.

In the asynch condition, the onset of sound presentation was based on the RR intervals extracted from a previously acquired synch block. Specifically, the ECG recorded during the preceding synch block was analyzed offline to extract RR intervals by automatic detection of R peaks and computation of RR intervals. 250 RR intervals were selected if they were above the 25th and below the 75th percentile of RR interval distribution in the synch block, in order to take into account possible missed R peaks in the online detection during the synch block. Next, RR interval order was shuffled giving rise to a predefined pseudo-random sequence closely resembling the participant's heartbeat rhythm. By construction, differences between the synch and asynch conditions in terms of average and variance of the RR intervals were minimized, contrary to the RS interval being fixed in the synch condition and variable in the asynch condition.

In the isoch condition, the onset of sound presentations was based on the median RR interval calculated during a previously acquired synch block. This procedure produced similar sound-to-sound intervals across the synch, asynch and isoch conditions however, unlike the synch and asynch conditions, sound-to-sound intervals in the isoch condition had low variability.

## Data acquisition

Continuous EEG (g.HIamp, g.tec medical engineering, Graz, Austria) was acquired at 1200 Hz from 63 active ring electrodes (g.LADYbird, g.tec medical engineering) arranged according to the international 10–10 system and referenced online to the right earlobe and offline to the left and right ear lobes. Electrode AFz served as the ground. Biophysical data were acquired using single-use Ag/AgCl electrodes. Three-lead ECG was recorded by attaching two electrodes (a third was a reference) to the participant's chest on the infraclavicular fossae and below the heart. A vertical EOG electrode was attached below the right eye and a horizontal EOG electrode was attached to the outer right canthus. Since muscle atonia is associated with increased sleep depth and is an essential marker for effective sleep staging[79], EMG was additionally acquired sub-mentally during the sleep session alone. Impedances of all active electrodes were kept below 50 kΩ. All electrophysiological data were acquired with an online band-pass filter between 0.1

and 140 Hz and a band-stop filter between 48 and 52 Hz to reduce electrical line noise.

## Sleep scoring

An experienced sleep scoring specialist (Somnox SRL, Belgium), blind to the experimental manipulation in this study, performed the scoring of the continuous sleep electrophysiological data in order to pinpoint the periods of wakefulness and micro-arousal as well as periods of N1, N2, N3 and REM sleep. Sleep scoring was performed via visual inspection of contiguous 30-second segments of the EEG, EOG and EMG time-series, as outlined in the 2007 AASM guidelines for sleep scoring[79]. Segments scored as periods of wakefulness or micro-arousals in the sleep recordings were excluded from further analysis.

## Data analysis

Electrophysiological data analyses were performed in MATLAB (R2019b, The MathWorks, Natick, MA) using open-source toolboxes EEGLAB (version 13.4.4b[80]), Fieldtrip (version 20201205[81]), as well as using custom-made scripts. Raincloud plots were generated using the Raincloud plot toolbox[82].

## R peak detection

The R peaks in the continuous raw ECG signal were selected offline using a semi-automated approach as in Pfeiffer & De Lucia[30]. The custom-made MATLAB script, *peakdetect.m* (https://ch.mathworks.com/matlabcentral/fileexchange/72-peakdetect-m/content/peakdetect.m), was utilized to automatically identify the sharp R peaks in the raw ECG signal. Visual inspection of the online and offline detected R peaks ensured that the selected peaks fitted within the expected structure of the QRS complex in the continuous raw ECG signal. Frequent flawed online identification of the R peaks or faulty auditory stimulus presentation in a given block resulted in the exclusion of the block from a given participant's dataset. For blocks that were included, unrealistic RR, RS and SR interval values, observed as a result of infrequent flawed offline marking of R peaks, were identified and excluded using the *rmoutliers* MATLAB function (R2019b, The Math-Works, Natick, MA) and visual inspection of the detected R peaks and selected outliers.

## Quality control analyses

A series of control analyses were performed to investigate whether the experimental manipulation was producing the expected RS, SR, RR and SS mean and variances, and the presence of possible confounding factors. SS and RR intervals (for sound trials preceded by sound trials) were extracted for the synch, asynch, isoch and baseline conditions where relevant. In addition, RS intervals for sound trials and SR intervals for omission trials were computed to quantify the degree of cardio-audio synchronization and heartbeat onset variability during sound and omission, respectively. Variability in the same interval measures, computed as the SEM was additionally investigated. Of note, unlike RR, SR and RS variability, SS variability was first computed within a given experimental block and then across experimental blocks, to account for ongoing changes in RR intervals (and hence SS intervals) in sleep. Non-parametric one way repeated measures Friedman tests ($p < 0.05$) were performed on the average RS, SR, SS and RR intervals and variabilities of each participant for wakefulness and all sleep stages with within-subject factor Condition (3 levels for RS, SR, SS intervals and variability: synch, asynch, isoch; 4 levels for RR intervals and variability: synch, asynch, isoch, baseline). Post-hoc paired Wilcoxon signed-rank tests ($p < 0.05$) identified any significant pairwise comparisons (no multiple comparisons correction was applied since pairwise differences were of interest).

To ensure no significant differences in the ECG signal across experimental conditions, ECG waveforms during omissions were extracted from ECG recordings between −100 ms and 500 ms relative to R peak onset, matching the EEG trial-based analysis (see below, EEG Data Analysis). The non-parametric cluster-based permutation statistical analysis approach[83] was employed to investigate ECG waveform differences between the various experimental conditions outlined herein. In order to reject the null hypothesis that no significant differences existed in the given set of experimental conditions being contrasted, maximum cluster-level statistics were determined by shuffling condition labels (5000 permutations), allowing for a chance-based distribution of maximal cluster-level statistics to be estimated. Since maxima are utilized by this method, it enables the correction of multiple comparisons over time. A two-tailed Monte-Carlo p-value allowed for the definition of a threshold of significance from the distribution ($p < 0.05$, two-tailed).

## ECG data analysis

The identified R peaks were used to derive omission trial related RR intervals. The representative R peak for an omission trial (the omission onset was obtained by adding the average SS interval within each block to the sound onset prior to the omission) was selected as the first R peak following an omission in the continuous ECG signal. The omission RR Interval ($RR_{om}$) was therefore calculated as the latency between the selected omission R peak and the R peak immediately preceding it. In order to investigate potential heartbeat alterations associated with the omission trial, additional RR intervals were identified using contiguous R peaks to reflect the RR interval for the trial prior to omission ($RR_{-1}$) and up to two trials following omission ($RR_{+1}$ and $RR_{+2}$). Omission trials were considered only if they were followed by at least two sound stimuli to ensure no overlap between investigated RR intervals. RR intervals were normalized by subject-wise division of each of the investigated average RR intervals ($RR_{om}$, $RR_{+1}$ and $RR_{+2}$) by the average $RR_{-1}$. This normalization resulted in $RR_{-1}$ intervals equal to 1 in each auditory condition, vigilance state, and participant, therefore $RR_{-1}$ was not considered in the statistical analysis (see below, ECG statistics and reproducibility).

The 30-second sleep stage labeled epochs were used to label omission trials for all auditory conditions. For the analysis of the ECG signals during the wakefulness and sleep sessions, we imposed a minimum of 30 artifact-free trials for the ECG data analysis and for each condition of interest. Hence, different sets of RR intervals formed the final dataset per participant and consisted of five vigilance states: AWAKE, N1, N2, N3 and REM sleep where available (Supplementary Tables 1, 2). One way ANOVAs on RR interval quantities showed no significant differences ($p > 0.05$) in trial numbers across auditory conditions within each vigilance state, therefore all RR intervals were included in the statistical analysis. RR intervals for each omission trial order type were averaged for each auditory condition, participant, and each vigilance state.

## ECG statistics and reproducibility

To investigate whether changes in normalized RR intervals could be explained by Auditory Condition (synch, asynch, isoch), Trial Order ($RR_{om}$, $RR_{+1}$ and $RR_{+2}$) or Vigilance State (AWAKE, N1, N2, N3, REM), we computed linear mixed-effects models using the *fitlme* function, as implemented in MATLAB (https://ch.mathworks.com/help/stats/fitlme.html). We generated the following two models with normalized RR interval as the dependent variable, Auditory Condition (synch, asynch, isoch), Trial Order ($RR_{om}$, $RR_{+1}$ and $RR_{+2}$) or Vigilance State (AWAKE, N1, N2, N3, REM) as fixed factors and subject as the random factor:

$$Model 1 : RR\ interval \sim Auditory\ Condition * Trial\ Order$$
$$+ Vigilance\ state * Auditory\ Condition$$
$$+ Vigilance\ state * Trial\ Order + (1|Subject)$$

$$Model 2 : RR\ interval \sim Auditory\ Condition$$
$$* Trial\ Order * Vigilance\ state + (1|Subject)$$

Post-hoc Wilcoxon signed-rank tests with Bonferroni correction for multiple comparisons ($p < 0.018$ for auditory condition and $p < 0.013$ for omission trial order comparisons) were utilized for pairwise comparisons between all investigated within-subject variables.

## EEG data analysis

Continuous raw EEG data were band-pass filtered using second-order Butterworth filters between 0.5 and 30 Hz for the wakefulness and sleep session. Independent Component Analysis (ICA) as implemented in Fieldtrip[80,81] was used in order to identify and remove any cardiac field and eye movement activity related components in the continuous wakefulness and sleep EEG.

Wakefulness and sleep continuous EEG data were epoched between −100 ms and 500 ms relative to the selected event onset. This range was chosen in reflection of previously reported classic omission response latencies ([12] and others) and importantly, to ensure no overlap between consecutive auditory stimulation trials (as in[30], assuming a range of 60–100 beats per minute across volunteers). Various event onsets were selected, namely, heartbeat onset, sound onset and omission onset, depending on the comparison performed (Results. Neural omission response). Of note, while the isoch condition had fixed SS intervals, this was not the case in the asynch condition. To test whether sound prediction was based on computing the time of maximal sound occurrence probability, average sound-based omission responses were calculated by extracting epochs from the continuous EEG asynch and isoch recordings, now time-locked to the latency occurring during the omission at the average SS interval and resulting in the estimation of the OEP.

The resting-state baseline recordings acquired as part of the experimental procedure (without auditory stimulation) were utilized to extract EEG evoked potentials with matched event onsets to auditory stimulation trials and epoched between −100 ms and 500 ms. First, we extracted HEPs based on heartbeat onset in the resting state for upcoming statistical comparisons to the synch and asynch OHEP. Second, in the absence of sound stimulation, the baseline allowed for comparing the OEPs of the isoch and asynch conditions to a control condition. In this case, a random selection of epochs was extracted from continuous baseline recordings, such that the latencies between epoch onset and closest heartbeat (i.e. R peak) were matched on the single-trial level to the trial onsets in the sound-based isoch and asynch conditions. Upcoming pre-processing steps were replicated for auditory stimulation and baseline trials, for each participant, and for wakefulness and all sleep stages as follows.

Artifact electrodes and trials were identified using a semi-automated approach for artifact rejection as implemented in Fieldtrip[81]. Noisy EEG electrodes were excluded based on a signal variance criterion (3 z-score Hurst exponent) and substituted with data interpolated from nearby channels using spherical splines[84]. Across eligible participants, an average of 8.0 (SD = 2.3) electrodes (M = 12.9%, SD = 3.7%) were interpolated in wakefulness and an average of 8.9 (SD = 1.1) electrodes (M = 14.4% SD = 1.7%) were interpolated in sleep. In wakefulness data, trials containing physiological artifacts (e.g. eye movement, excessive muscle activity), not accounted for by ICA, were identified by visual inspection and by using a 70 μV absolute value cut-off applied to the EEG signal amplitude and were excluded from further analysis. A higher absolute value cut-off of 300 μV was employed for the selection of artifactual epochs in sleep, in order to prevent the exclusion of high-amplitude slow wave activity[85]. Extreme outliers in signal kurtosis and variance were additional criteria for the rejection of artifactual trials from sleep recordings. Finally, common average re-referencing was applied.

The 30-second sleep stage labeled epochs were used to label artifact-free trials for all event onset types and for all three auditory conditions and the baseline. Hence, five different sets of trials formed the final processed dataset per participant for each of the five vigilance states: AWAKE, N1, N2, N3 and REM sleep where available (Supplementary Tables 1, 2). We note that we chose not to employ pre-stimulus baseline correction in wakefulness or sleep trials. In addition, since one way ANOVAs on artifact-free trial numbers for each evoked response of interest confirmed that no significant differences ($p > 0.05$) existed across auditory conditions alone, all trials were kept in auditory condition comparisons. Conversely, since acquired baseline data were significantly less ($p < 0.05$) than the three auditory conditions, baseline trial numbers were quantitatively matched for each participant and for each vigilance state between auditory conditions and baseline.

The number of available omission trials for each vigilance state, averaged across experimental conditions and event onsets of interest were AWAKE: M = 286, SD = 13 trials; N1: M = 97, SD = 22 trials; N2: M = 428, SD = 184 trials; N3: M = 159, SD = 72 trials; REM: M = 163, SD = 65 trials. However, the sample size estimation (see Methods, sample size estimation) revealed that more than 17 participants were required for the comparison of synch *vs* asynch based on previous results. Therefore, we performed omission comparisons only for wakefulness and N2 sleep ($N = 23$; Supplementary Tables 1, 2). For asynch OHEPs, we additionally excluded trials where the previous sound occurred less than 500 ms before the R peak onset during the omission. Trial numbers were then quantitatively matched for the synch and baseline OHEP trials, to ensure a well-balanced number of trials on any given comparison of interest (see Supplementary Table 3 for condition-specific trial numbers). Finally, grand average evoked responses to sound omissions were derived.

## EEG statistics and reproducibility

For the analysis of the electrophysiological signals during the wakefulness and sleep sessions, we imposed a minimum of 60 artifact-free trials for the EEG data analysis, chosen based on the signal-to-noise ratio required to meaningfully interpret EEG statistical analysis results[86].

The non-parametric cluster-based permutation statistical analysis approach[83] was employed to investigate sensor-level EEG-based differences between the various experimental conditions outlined herein. Under this statistical framework, statistically significant individual data samples were grouped based on the degree of their shared spatial and temporal characteristics. The resulting clusters were statistically evaluated by summating the t-values for all samples forming up a given cluster. In order to reject the null hypothesis that no significant differences existed in the given set of experimental conditions being contrasted, maximum cluster-level statistics were determined by shuffling condition labels (5000 permutations), allowing for a chance-based distribution of maximal cluster-level statistics to be estimated. A two-tailed Monte-Carlo p-value allowed for the definition of a threshold of significance from the distribution ($p < 0.05$, two-tailed). Of note, the cluster permutation based multiple comparisons correction only applied across channels and latencies when comparing two experimental conditions, however no multiple comparisons correction was applied across the number of comparisons made in this study. In wakefulness and sleep, this procedure was performed over the entire trial length from −100 ms to 500 ms relative to the event onset of interest (heartbeat onset for OHEP comparisons and expected sound onset based on average SS interval for OEP comparisons). Finally, in order to evaluate the size of the observed effects, the Cohen's d statistic was calculated at the peak latenies of significant clusters[87].

## SO data analysis

To examine whether N2 sleep EEG statistical analysis results could have been influenced by the relationship between sounds and SO activity[40,42,69] and/or between heartbeats and SO activity[33] in NREM sleep, we identified SOs in N2 sleep artifact-free EEG data. We then computed the latency at which the SOs occurred compared first, to sound presentations in the auditory conditions and second, to heartbeats in the auditory conditions and baseline.

SOs in the continuous EEG time-series were detected across experimental conditions over frontocentral electrode Cz where a high probability for SO detection is to be expected[67]. We marked SOs based on the method described in Ngo et al.[42] and in Besedovsky et al.[69]. In brief, the 0.5 to 30 Hz band-pass filtered data were downsampled from 1200 Hz to 100 Hz and a low pass finite impulse response filter of 3.5 Hz was used in order to improve the detection of SO components. Next, N2 sleep labeled segments were extracted for the identification of SOs. Consecutive positive-to-negative zero crossings were picked out and were selected only if their temporal distance was between 0.833 s and 2 s, yielding a frequency between 0.5 Hz and 1.2 Hz for the designated oscillations. Negative and positive peak potentials in the oscillations were defined as the minima and maxima present within eligible

consecutive positive-to-negative zero crossings. The mean negative and mean positive-to-negative amplitude differences were calculated across selected oscillations at electrode Cz. For each identified oscillation that satisfied the frequency criterion, if the negative amplitude was 1.25 times lower than the mean negative amplitude and the positive to negative amplitude difference was 1.25 times higher than the mean positive to negative amplitude difference, the oscillation was marked as a SO.

The positive half-wave peak time point was chosen as the representative latency for each SO in light of relevant literature[68,70] and following visual inspection of the SO and sound presentation time-series. Sound to SO latency for the synch, asynch and isoch conditions and R peak to SO latency for the synch, asynch, isoch and baseline conditions were computed for all artifact-free sound trials. Omission trials were excluded from this analysis since in this case, we were interested in how sounds and the sound to heartbeat relationship may modulate SOs in N2 sleep. Latencies were considered as valid only if they were between −800 and 800 ms[33], a range chosen to minimize potential contamination of the sound or R peak to SO relationship by upcoming sounds or heartbeats. Median latencies were calculated for each subject and condition separately.

## SO statistics and reproducibility

In N2 sleep, a non-parametric $1 \times 3$ repeated measures Friedman test was calculated on the median sound to SO latencies at electrode Cz with within-subject factor Auditory Condition (synch, asynch, isoch) and a $1 \times 4$ repeated measures Friedman test was computed on the median heartbeat to SO latencies with within-subject factor Condition (synch, asynch, isoch, baseline). Pairwise comparisons were calculated using post-hoc paired Wilcoxon signed-rank tests between all investigated within-subject variables (no multiple comparisons correction was applied since pairwise differences were of interest). Latency trial numbers were matched for the median heartbeat to SO latency comparison but not for the sound to SO latency comparison, since in the former, baseline recordings lengths were significantly lower to auditory condition recordings (as confirmed by $1 \times 4$ and $1 \times 3$ repeated measures ANOVAs on latency trial numbers, which significantly differed only across heartbeat to SO trials and not sound to SO trials).

## Data availability

All data supporting the findings of this study are provided in an OSF public repository (DOI: 10.17605/OSF.IO/HMQZA)[88].

## Code availability

Costum-made code used to run the quality control analyses and EEG and ECG data analyses are available on a GitHub public repository (https://github.com/DNC-EEG-platform/CardioAudio_Sleep/). Any additional information required to reanalyze the data reported in this paper is available from the corresponding authors upon request.

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

## Acknowledgements

We express our gratitude to Nathalie Nguepnjo Nguissi, Elsa Dosi and Diana Ortolani for assistance during data acquisition. We extend our thanks to Dr Aurore Guyon Postalci from SOMNOX for performing the stage scoring of the sleep data. We thank Rupert Ortner and the g.tec team for technical support. This work was supported by a Spark SNSF Grant (no. CRSK-3_196194), the SNSF Grant (no. 32003B_212981) and the Catalyst Fund of the Bertarelli Foundation awarded to M.D.L. Additional funding was provided by an SNSF Grant (no. 320030_182589) awarded to S.S.

## Author contributions

A.P., C.P., S.S. and M.D.L. designed the experiment and conceived the analysis. A.P. conducted the experiment and acquired the data. A.P. and M.D.L. analyzed the data and wrote the manuscript. C.P. and S.S. edited the manuscript.

## Competing interests

The authors declare no competing interests.
