## [Peer review file · Communications Biology]

Reviewers' comments:

Reviewer #1 (Remarks to the Author):

The authors observe interesting differences in omission responses when comparing sound sequences related and unrelated to the heartbeat across vigilance states. They present a cardiac deceleration response post-omission in synchronous cardiac-auditory sequences, which isn't present in isochronous auditory sequences and asynchronous cardiac-auditory sequences. This is present across all vigilance states. Furthermore, they observe differences in omission heartbeat evoked potentials (HEPs) when comparing synchronous cardiac-auditory sequences with asynchronous cardiac-auditory sequences, as well as differences in omission evoked potentials (OEPs) across isochronous sound sequences and asynchronous cardiac-auditory sequences. These differences are distinct in wakefulness than in N2 sleep.

Comments:

1. you state the asynchronous condition was based on a previous synchronous trial. Did you check that the sounds were not synchronised to the present heartbeat sequences for asynchronous trials?

2. Have you considered whether a 52 second delay from the heartbeat is too short for a prediction of a sound from the heartbeat?

3. Did you correct for cardiac field artefact? It is recommended to correct for this when analysing HEPs as ERP differences may reflect differences in cardiac activity/cardiac field artefacts, rather than cognitive differences (Park & Blanke., 2019).

4. As a control, it could be important to check how close ECG R-peaks are to omission evoked responses (your analysis window) across conditions. As well as removing trials with sounds too close to the asynchronous omission HEP response. Also for your auditory evoked potentials. Differences in the position of R-peaks and sounds across conditions may confound results.

5. mild correction:  there are 2 'between between' at line 422

Thank you for your article.

REFERENCES:

Park, H. D., & Blanke, O. (2019). Heartbeat-evoked cortical responses: Underlying mechanisms, functional roles, and methodological considerations. *Neuroimage*, 197, 502-511.

Reviewer #2 (Remarks to the Author):

In this work, Pelentritou and colleagues investigated during sleep and wakefulness the ECG and EEG changes induced by unexpected sound omission in three different conditions: sounds synchronous with the heartbeat, isochronous, and asynchronous. They found that sound omission was associated with a heartbeat deceleration in all vigilance states. Moreover, they observed that auditory regularities induced prediction of upcoming sounds both when sounds occurred at a fixed frequency and when temporally synchronized to the ongoing heartbeat during wakefulness and N2 sleep. Finally, they showed that sound regularity was associated with a reorganization of slow wave activity.

The study is novel and interesting. In my opinion, though, the rationale behind the different analyses and the meaning of the different results are not clearly presented and explained. Given that this is a complex study with many different conditions and analyses, this lack of clarity makes it very difficult for the reader to follow the authors' reasoning. Below are my further comments and suggestions for the authors.

Lines 158-161: "Permutation testing followed by Wilcoxon signed-rank tests evaluated the distribution of post-permutation F values against the original F values and confirmed the significance of the main effects and interactions ($p < 0.0005$)". Even after reading the Methods section, I am not sure about what the authors did here. The text seems to suggest that the null distribution was directly entered into statistical tests that compared it with the actual results. However, this could be an issue as the number of permutations would affect the DoFs and statistical power. Please clarify.

Lines 170-175: "To investigate whether the heartbeat deceleration was modulated by vigilance state and auditory conditions (Figure 2B), we computed a three-way repeated measures ANOVA with factors Vigilance State (AWAKE, N1, N2, N3, REM), Auditory Condition (synch, asynch, isoch), and Trial Order (one trial before, trial during, first trial after, second trial after sound omission), including only participants who had sufficient data in all vigilance states (N=6; Supplementary Table 2)". The included sample size seems very small for an interpretable three-way repeated measures ANOVA. Are there any specific reasons for not using more sophisticated statistical models allowing to handle missing data values and thus to include the full sample?

Line 296-299: "As outlined above (Figure 4B), the analysis of OHEPs during N2 sleep (N=23) revealed that cardio-audio synchronization (compared to the asynch condition) gave rise to a central early onset positivity (at -99 ms to 117 ms) and late onset negativity (at 322 ms to 500 ms). This effect is reminiscent of the up and down states of slow oscillations (SOs; 0.5-1.2 Hz oscillations) during N2 sleep".

Figure 5: What are panels A and B supposed to show? The two panels do not seem to show any clear slow-wave-like signal change (besides maybe for the synchronous condition). In my opinion, the whole analysis of sleep slow waves and its relationship with the other analyses in the study is not very clear. Some figures or schemes could help the reader understand what has been done. Please also show the mean signal profile of detected slow waves or some representative traces with marked detections.

Lines 837-840: "The number of available trials for each vigilance state, averaged across experimental conditions and event onsets of interest were AWAKE: M = 286, SD = 13 trials; N1: M = 97, SD = 22 trials; N2: M = 428, SD = 184 trials; N3: M = 159, SD = 72 trials; REM: M = 163, SD = 65 trials". Please also indicate in text or (supplementary) table the number of trials included per experimental condition.

Please consider adding color legends in figures to improve readability.

Abbreviations such as RS, SS, SR, and RR are spelled out and explained very late in the manuscript. This should be done earlier.

Reviewer #1 (Remarks to the Author):

The authors observe interesting differences in omission responses when comparing sound sequences related and unrelated to the heartbeat across vigilance states. They present a cardiac deceleration response post-omission in synchronous cardiac-auditory sequences, which isn't present in isochronous auditory sequences and asynchronous cardiac-auditory sequences. This is present across all vigilance states. Furthermore, they observe differences in omission heartbeat evoked potentials (HEPs) when comparing synchronous cardiac-auditory sequences with asynchronous cardiac-auditory sequences, as well as differences in omission evoked potentials (OEPs) across isochronous sound sequences and asynchronous cardiac-auditory sequences. These differences are distinct in wakefulness than in N2 sleep.

We thank the reviewer for the thorough consideration of our manuscript and the constructive feedback. Below, we addressed all the raised comments. Changes in the manuscript and supplementary materials are outlined herein and can also be identified in the revised manuscript and supplementary materials in red.

Comments:

1. you state the asynchronous condition was based on a previous synchronous trial. Did you check that the sounds were not synchronised to the present heartbeat sequences for asynchronous trials?

We thank the reviewer for raising this important point. Yes, we checked that in the asynchronous (asynch) trials, sound onsets were not synchronized to the heartbeat. This check was based on the control analysis investigating RS intervals and their variability. In the synchronous (synch) condition, we found values of the average RS variability (Standard Error of the Mean of RS intervals) ranging between 0.1 ms and 0.2 ms across wakefulness and all sleep stages. In the asynch condition these values were much higher and ranging between 6.8 ms and 14.9 ms in wakefulness and all sleep stages. These values were significantly different in the synch compared to the asynch (or isochronous (isoch)) condition as outlined in text (*page 17, lines: 326-331*). The group and single subject values across wakefulness and all stages of sleep are shown in *Supplementary Figure 3A,B* with the group values now added to the manuscript (*pages 16-17, lines 321-326*) to further clarify the differences in RS variability across conditions as follows:

'In the synch condition, the average RS interval was 52.3 ms (SEM = 0.1 ms) for sound trials, and -2.6 ms (SEM = 2.6 ms) for the isoch and asynch conditions across wakefulness and all sleep stages (Supplementary Figure 3A). We observed the expected lower RS interval variability in the synch relative to the two other conditions (isoch and asynch) with values of

0.2 ms (SEM = 0 ms) for the synch condition and 10.6 ms (SEM = 0.7 ms) for the isoch and asynch conditions across wakefulness and all sleep stages.'

Here, as an example, we also display all RS intervals for an exemplar subject in wakefulness (Figure R1 in this letter), with similar results obtained across all volunteers and across wakefulness and all sleep stages. We conclude that, in the asynch condition there was no fixed time interval between heartbeats and sound onsets.

Figure R1. RS intervals in one exemplar participant during wakefulness demonstrating the absence of synchronization to the present heartbeat in the asynchronous condition (asynch, black). The experimentally imposed ~50 ms delay between R peak and Sound in the synchronous condition (synch, red) is also clearly visible. The RS variability (Standard Error of the Mean of RS intervals) for this subject was 0.1 ms for the synch condition and 4.7 ms for the asynch condition.

2. Have you considered whether a 52 second delay from the heartbeat is too short for a prediction of a sound from the heartbeat?

The reviewer raises an important point, which indeed requires clarification. We expected that the auditory prediction (if any) in the synch condition would be formed based on a series of sounds presented at a certain timing relative to the heartbeat cycle (here presented 52 ms after the R peak). This was essentially our experimental question to be tested and our results show that the delay of 52 ms between R peak and sound onset in the synch condition is sufficient for the brain to anticipate upcoming sounds based on the timing of the ongoing heartbeat, evident in both the ECG and EEG omission responses. We cannot of course exclude that these omission responses result from cardiac-related information conveyed to the brain at latencies that precede the R peak. Indeed, by fixing the R peak-to-sound onset time

interval, we implicitly impose a stable temporal relationship between the heartbeat cycle as a whole and the administered sounds. If so, we are not restricting the investigation of the cardio-audio inference to the short delay of 52 ms, but rather to latencies that can range up to the average RR intervals. We have now clarified this point in the revised manuscript on *page: 23, lines: 507-513* as follows:

'In addition, the current results do not allow for a strict identification of the latency at which the cardiac and neural signals are integrated at the neural level. Indeed, by imposing a fixed R peak-to-sound interval in the synch condition, we indirectly impose a stable temporal relationship between any point within the heartbeat cycle - due to its relatively fixed periodicity - and the sound onset. Considering this, we cannot exclude that the ECG and EEG omission responses may arise from cardiac related information conveyed to the brain at earlier latencies than the R peak.'

3. Did you correct for cardiac field artefact? It is recommended to correct for this when analysing HEPs as ERP differences may reflect differences in cardiac activity/cardiac field artefacts, rather than cognitive differences (Park & Blanke., 2019).

We thank the reviewer for this useful suggestion. Following the reviewer's advice, we have now corrected for the cardiac field artefact in the wakefulness and sleep EEG data by removing cardiac components via Independent Component Analysis (as recommended in Park & Blanke, 2019). For wakefulness, other than the expected reduction in the amplitude of the first high-amplitude component of the heartbeat evoked potential (HEP) during omissions, the results of the cluster-permutation statistical analysis in the omission HEPs (OHEPs) were highly similar (see Table R1 in this letter for the AWAKE synch vs asynch comparison). This suggests that differences between omission responses in the synch and isoch conditions compared to the asynch and/or baseline condition were independent of the neural response to the heartbeat signal itself.

Removing the cardiac field artefact in sleep EEG data based on the removal of independent components containing the cardiac artifacts resulted in different findings in N2 sleep. More specifically, the early cluster at the R peak onset when contrasting the synch and asynch omission responses was no longer significant (Table R1 in this letter). Nonetheless, the late cluster was still present despite the removal of the cardiac field artefact, suggesting that indeed a prediction was formed based on the relationship between R peaks and sounds in the synch condition in N2 sleep (Table R1). In addition, for N2 sleep, we identified a new negative cluster in the isoch vs baseline OEP comparison (new *Supplementary Figure 2*), further pointing to deviance detection upon omission in classic auditory regularity processing.

Figures 3 and 4 and Supplementary Figure 1 have now been updated to demonstrate the results following correction for the cardiac field artefact. A new figure (Supplementary Figure 2) has been added for the differential response between isoch and baseline OEPs in N2 sleep. The updated results can be found in the manuscript (page 11, lines 213-225; pages 13-14, lines 254-276) and the methods have been updated as follows (page 34, lines 777-780):

'Independent Component Analysis (ICA) as implemented in Fieldtrip^{80,81} was used in order to identify and remove any cardiac field and eye movement activity related components in the continuous wakefulness and sleep EEG.'

4. As a control, it could be important to check how close ECG R-peaks are to omission evoked responses (your analysis window) across conditions. As well as removing trials with sounds too close to the asynchronous omission HEP response. Also for your auditory evoked potentials. Differences in the position of R-peaks and sounds across conditions may confound results.

We thank the reviewer for this constructive comment with regards to the interval between the previous sounds and the R peaks during omissions in the asynch condition. To that effect, we performed the synch vs asynch omission heartbeat evoked potentials (OHEPs) comparison in wakefulness and N2 sleep after removing trials for which a sound onset was closer than 500 ms to the R peak of an omission trial. In the synch condition, by construction, there were no trials during omissions for which the previous sound trial was closer than 500 ms to the R peak during omission. In the asynch condition, this procedure led to the exclusion of 19% of trials for wakefulness (number of excluded trials: Mean \pm Standard Deviation = 53 \pm 25) and 10% of trials for N2 sleep (number of excluded trials: Mean \pm Standard Deviation = 40 \pm 27) (Table R1 in this letter). To ensure a similar number of trials across conditions for the statistical comparisons, we then randomly excluded the required number of trials from synch OHEP and baseline HEP trials. The cluster permutation statistical analysis following this exclusion produced highly similar results in comparison to what previously obtained, *i.e.*, for those differential responses that survived after the exclusion of the independent component related to the cardiac artifact (see response to Reviewer #1, comment 3 and Table R1 in this letter).

Figure 4 and Supplementary Figure 1 have now been updated to illustrate the new results following exclusion of the aforementioned trials. The updated results can be found in the manuscript (page 11, lines 213-225) and the methods have been updated as follows (page 36, lines 834-838):

'For asynch OHEPs, we additionally excluded trials where the previous sound occurred less than 500 ms before the R peak onset during the omission. Trial numbers were then quantitatively matched for the synch and baseline OHEP trials, to ensure a well-balanced

number of trials on any given comparison of interest (see Supplementary Table 3 for condition-specific trial numbers).’

Regarding the reviewer’s second comment on auditory evoked potentials (AEPs), which are only statistically contrasted in the isoch and asynch (to verify no differences in auditory stimulus processing as outlined on page 9, lines 191-193), we compared the average RS and SR intervals between isoch and asynch and we found no statistically significant differences (Results 5.1. Experimental paradigm control analyses; Figures 3A,C). This suggests no differences in the R peak to Sound relationship between the isoch and asynch conditions. The relationship between R peak and Sound would be problematic if we compared the synch AEPs to those of the isoch and asynch condition, which we refrained from testing as the synch condition AEP is superimposed with the HEP (as explained on page 9, lines 193-195). In addition, as per the reviewer’s recommendation, we have now removed the cardiac field artefact using independent component analysis prior to the calculation of the AEPs (see response to Reviewer #1, comment 3) therefore, the impact of the cardiac field on the AEPs should now be minimized.

Table R1. Previous (OLD) and updated (NEW) results for the OHEP and OEP comparisons performed in this study following cardiac field artefact correction (using Independent Component Analysis) and removal of OHEP trials with sounds too close to the asynch OHEP (<500 ms distance between previous sound and omission trial).

Comparison	Trial Numbers (Mean ± STD)		Clusters		Cluster Periods (ms)		Effect Size (Cohen’s d)	
	OLD	NEW	OLD	NEW	OLD	NEW	OLD	NEW
AWAKE: synch vs asynch OHEP	synch: 290 ± 8 asynch: 284 ± 18	synch: 231 ± 29 asynch: 231 ± 29	negative (p=0.033)	negative (p=0.027)	165-218	158-216	1.055	0.841
N2 SLEEP: synch vs asynch OHEP	synch: 437 ± 191 asynch: 427 ± 181	synch: 368 ± 167 asynch: 388 ± 167	negative (p=0.017)	negative (p=0.044)	322-500 -99-117	332-500	0.622 0.836	0.762

5. mild correction: there are 2 ‘between between’ at line 422

We thank the reviewer for identifying this typo which has now been corrected.

Thank you for your article.

REFERENCES:

Park, H. D., & Blanke, O. (2019). Heartbeat-evoked cortical responses: Underlying mechanisms, functional roles, and methodological considerations. *Neuroimage*, 197, 502-511.

Reviewer #2 (Remarks to the Author):

In this work, Pelentritou and colleagues investigated during sleep and wakefulness the ECG and EEG changes induced by unexpected sound omission in three different conditions: sounds synchronous with the heartbeat, isochronous, and asynchronous. They found that sound omission was associated with a heartbeat deceleration in all vigilance states. Moreover, they observed that auditory regularities induced prediction of upcoming sounds both when sounds occurred at a fixed frequency and when temporally synchronized to the ongoing heartbeat during wakefulness and N2 sleep. Finally, they showed that sound regularity was associated with a reorganization of slow wave activity.

The study is novel and interesting. In my opinion, though, the rationale behind the different analyses and the meaning of the different results are not clearly presented and explained. Given that this is a complex study with many different conditions and analyses, this lack of clarity makes it very difficult for the reader to follow the authors' reasoning. Below are my further comments and suggestions for the authors.

We thank the reviewer for his/her encouraging comments and constructive feedback. Below, we addressed all the raised comments. Changes in the manuscript and supplementary materials are outlined herein and can also be identified in the revised manuscript and supplementary materials in red.

In this revision, we have simplified and replaced some of the previously presented analyses and results following the reviewers' suggestions. In addition, we have attempted to simplify the text and clearly outline the motivation behind each of the analyses throughout the manuscript. We believe that the manuscript is now much clearer (for example, we have added a new paragraph clearly outlining the reasoning and performed analyses at the beginning of the Results section, see *pages: 5-6, lines: 104-117*).

In addition to the point-by-point responses that we present below, it should be noted that the manuscript has been also modified following the other reviewer's suggestions. This implies that the formulation of some of the raised comments may no longer apply to the current stage of the study. For example, please note that, following the removal of independent components related to the cardiac artifacts, the OHEPs comparison during N2 sleep provided significant results only during the 332-500ms post R peak period.

Lines 158-161: "Permutation testing followed by Wilcoxon signed-rank tests evaluated the distribution of post-permutation F values against the original F values and confirmed the significance of the main effects and interactions ($p < 0.0005$)". Even after

reading the Methods section, I am not sure about what the authors did here. The text seems to suggest that the null distribution was directly entered into statistical tests that compared it with the actual results. However, this could be an issue as the number of permutations would affect the DoFs and statistical power. Please clarify.

As per the reviewer's suggestion in another comment, we have now replaced both the two-way and three-way ANOVA analyses and subsequent permutation testing with more sophisticated linear mixed models (see below in the next reviewer suggestion's response). Therefore, sections describing the permutation testing have been removed.

The new results using linear mixed models are described in *pages 7-8, lines 139-166* and the new methods have been updated on *page 33, lines 762-771* as follows:

'To investigate whether changes in normalized RR intervals could be explained Auditory Condition (synch, asynch, isoch), Trial Order (RR_{om} , RR_{+1} and RR_{+2}) or Vigilance State (AWAKE, N1, N2, N3, REM), we computed linear mixed-effects models using the fitlme function, as implemented in MATLAB (<https://ch.mathworks.com/help/stats/fitlme.html>). We generated the following two models with normalized RR interval as the dependent variable, Auditory Condition (synch, asynch, isoch), Trial Order (RR_{om} , RR_{+1} and RR_{+2}) or Vigilance State (AWAKE, N1, N2, N3, REM) as fixed factors and subject as the random factor:

*Model 1: RR interval ~ Auditory Condition*Trial Order + Vigilance state*Auditory Condition + Vigilance state*Trial Order + (1|Subject)*

*Model 2: RR interval ~ Auditory Condition*Trial Order* Vigilance state + (1|Subject)'*

Lines 170-175: "To investigate whether the heartbeat deceleration was modulated by vigilance state and auditory conditions (Figure 2B), we computed a three-way repeated measures ANOVA with factors Vigilance State (AWAKE, N1, N2, N3, REM), Auditory Condition (synch, asynch, isoch), and Trial Order (one trial before, trial during, first trial after, second trial after sound omission), including only participants who had sufficient data in all vigilance states (N=6; Supplementary Table 2)". The included sample size seems very small for an interpretable three-way repeated measures ANOVA. Are there any specific reasons for not using more sophisticated statistical models allowing to handle missing data values and thus to include the full sample?

We thank the reviewer for this constructive comment. With that in mind, we replaced the two-way and three-way ANOVA analyses and investigated whether the normalized interbeat intervals (RR) could be explained by Vigilance State, Auditory Condition and Trial Order using linear mixed-effects models, which allow for missing values across subjects (which served as our random factor). The model comparing the single effects and interactions corroborated our repeated measures ANOVA results for Auditory Condition and Trial Order within each vigilance

state, since both the single terms and the interaction term were significant. Conversely, Vigilance State and the interaction terms of Vigilance State x Auditory Condition as well as Vigilance State x Trial Order did not significantly explain the normalized RR values suggesting no effect of Vigilance state on the observed heartrate deceleration upon omission. These results were corroborated by the absence of the significant triple interaction effect for Vigilance State x Auditory Condition x Trial Order.

In agreement with the reviewer, we have now removed *Figure 2B* on the reduced cohort of N=6, as the small sample size was not providing additive information to *Figure 2A* (updated *Figure 2*). The new results using linear mixed models are described in *pages 7-8, lines 139-166 and the new methods have been updated on page 33, lines 762-771*.

Line 296-299: “As outlined above (Figure 4B), the analysis of OHEPs during N2 sleep (N=23) revealed that cardio-audio synchronization (compared to the asynch condition) gave rise to a central early onset positivity (at -99 ms to 117 ms) and late onset negativity (at 322 ms to 500 ms). This effect is reminiscent of the up and down states of slow oscillations (SOs; 0.5-1.2 Hz oscillations) during N2 sleep”.

Figure 5: What are panels A and B supposed to show? The two panels do not seem to show any clear slow-wave-like signal change (besides maybe for the synchronous condition). In my opinion, the whole analysis of sleep slow waves and its relationship with the other analyses in the study is not very clear. Some figures or schemes could help the reader understand what has been done. Please also show the mean signal profile of detected slow waves or some representative traces with marked detections.

We agree with the reviewer and have now attempted to be more explicit on the reasoning behind the slow oscillation analysis (*page 5, lines 110-113*) as follows:

‘In section 4, we explore how different types of auditory regularities affect the background slow oscillations (SOs) in N2 sleep, as one potential mechanism by which the sleeping brain might build predictions based on the temporal relationship between auditory stimuli.’

In addition, as per the reviewer’s suggestion, we have now included a brief explanation of the steps in the Results section (*pages 14-15, lines: 281-288*) as follows:

‘Since sound presentations are known to alter the background oscillatory activity in sleep, notably the SOs (0.5-1.2 Hz) during NREM sleep^{40,42,43}, we investigated whether the three auditory conditions had differential effects on the ongoing SOs during N2 sleep. After extracting the SOs during N2 sleep for each experimental condition at electrode Cz (Figure 5A), we identified the closest positive peak latency of each SO to a given sound or R peak (Figure 5B) and computed the median sound-to-SO latency for all auditory conditions and median R peak-

to-SO latency for all auditory conditions and the baseline (Figure 5C) for latencies between -800 ms and 800 ms³³.’

A thorough explanation of the steps taken to extract the slow oscillations and statistically assess their temporal relationship to sounds is additionally offered in *Methods. SO data analysis* (pages: 37-38, lines 862-893) and in *Methods. SO statistics and reproducibility* (page 38, lines: 894-906), respectively. In agreement with the reviewer, we also added new panels in Figure 5 (new Figure 5A with two sub-panels), where we show a representative EEG signal at electrode Cz for one subject with the detected peaks and troughs of slow oscillations as well as the mean signal for detected slow oscillations in the same subject. We additionally included Figure sub-legends (Figure 5A,B,C) that we believe further clarify the procedure followed for this analysis.

Lines 837-840: “The number of available trials for each vigilance state, averaged across experimental conditions and event onsets of interest were AWAKE: M = 286, SD = 13 trials; N1: M = 97, SD = 22 trials; N2: M = 428, SD = 184 trials; N3: M = 159, SD = 72 trials; REM: M = 163, SD = 65 trials”. Please also indicate in text or (supplementary) table the number of trials included per experimental condition.

We now include a supplementary table (*Supplementary Table 3*) reporting the number of trials available in each experimental condition for wakefulness and N2 sleep, for which the EEG statistical analyses were performed.

Please consider adding color legends in figures to improve readability.

We have added color legends in *Figure 2*, for which legends were missing. The remaining figures already had color legends.

Abbreviations such as RS, SS, SR, and RR are spelled out and explained very late in the manuscript. This should be done earlier.

We have now included a sentence specifying what these abbreviations refer to at the beginning of the Results section of the manuscript on *page: 6, lines: 118-121* as follows:

‘Note that in the following, R refers to the R peak of the ECG waveform, S represents the sound onset, RR the R peak-to-R peak time interval, RS the R peak-to-sound onset time interval, SR the sound to the next R peak time interval, SS the sound-to-sound time interval, and that the variability of these variables is calculated as the standard error of the mean (SEM).’

REVIEWERS' COMMENTS:

Reviewer #1 (Remarks to the Author):

The authors have addressed my comments, as well as added clarification in the edited version of the article. I appreciate the re-analysis of the OHEP and OEP results with cardiac field correction and removal of trials with potential confounding effects. The table R1 provided was very useful for noting differences in results after re-analysis. Thank you for your article.

Reviewer #2 (Remarks to the Author):

The authors adequately addressed all my comments. The applied revisions significantly improved the overall quality of the manuscript.